# On the Complexity of Finding Stationary Points in Nonconvex Simple Bilevel Optimization

**Jincheng Cao**
UT Austin
jinchengcao@utexas.edu

**Ruichen Jiang**
UT Austin
rjiang@utexas.edu

**Erfan Yazdandoost Hamedani**
The University of Arizona
erfany@arizona.edu

**Aryan Mokhtari**
UT Austin & Google Research
mokhtari@austin.utexas.edu

## Abstract

In this paper, we study the problem of solving a simple bilevel optimization problem, where the upper-level objective is minimized over the solution set of the lower-level problem. We focus on the general setting in which both the upper- and lower-level objectives are smooth but potentially nonconvex. Due to the absence of additional structural assumptions for the lower-level objective—such as convexity or the Polyak–Łojasiewicz (PL) condition—guaranteeing global optimality is generally intractable. Instead, we introduce a suitable notion of stationarity for this class of problems and aim to design a first-order algorithm that finds such stationary points in polynomial time. Intuitively, stationarity in this setting means the upper-level objective cannot be substantially improved locally without causing a larger deterioration in the lower-level objective. To this end, we show that a simple and implementable variant of the dynamic barrier gradient descent (DBGD) framework can effectively solve the considered nonconvex simple bilevel problems up to stationarity. Specifically, to reach an $(\epsilon_f, \epsilon_g)$-stationary point—where $\epsilon_f$ and $\epsilon_g$ denote the target stationarity accuracies for the upper- and lower-level objectives, respectively—the considered method achieves a complexity of $\mathcal{O}(\max(\epsilon_f^{-\frac{3+p}{1+p}}, \epsilon_g^{-\frac{3+p}{2}}))$, where $p \geq 0$ is an arbitrary constant balancing the terms. To the best of our knowledge, this is the first complexity result for a discrete-time algorithm that guarantees joint stationarity for both levels in general nonconvex simple bilevel problems.

## 1 Introduction

In this paper, we consider the following nonconvex simple bilevel optimization problem

$$\min_{\mathbf{x} \in \mathbb{R}^n} \ f(\mathbf{x}) \qquad \text{s.t.} \quad \mathbf{x} \in \mathcal{X}_g^* = \operatorname*{argmin}_{\mathbf{z} \in \mathbb{R}^n} \ g(\mathbf{z}), \tag{1}$$

where $f, g : \mathbb{R}^n \to \mathbb{R}$ are continuously differentiable and $\mathcal{X}_g^*$ denotes the solution set of the lower-level problem. This problem is referred to as simple bilevel. The term "simple" distinguishes this setting from general bilevel optimization, where the lower-level solution set $\mathcal{X}_g^*$ may depend on the upper-level variable, introducing additional complexity. Owing to its numerous applications in areas such as lifelong learning [1, 2] and over-parameterized machine learning [3, 4], simple bilevel optimization has garnered significant recent interest in understanding its structure and developing efficient algorithms for finding its solution [2, 4–6].

39th Conference on Neural Information Processing Systems (NeurIPS 2025).

The main challenge in solving Problem (1) stems from the fact that the feasible set, defined as the optimal solution set of the lower-level problem, lacks a clear characterization and is not explicitly given. As a result, a direct application of projection-based or projection-free methods is infeasible. Several works have studied the case where both the upper- and lower-level objectives are convex. In this case, Problem (1) is "well-behaved", facilitating the application of various optimization techniques. For instance, several works [7–10] employ Tikhonov regularization [11], combining the upper- and lower-level objectives with an appropriately chosen weight. Another line of research [2, 4, 12] provides a linear approximation of the lower-level objective to form an outer approximation of the lower-level optimal solution set $\mathcal{X}_g^*$.

However, in several applications such as neural network training [5], sparse representation learning [3, 6], and adversarial training [13–15], the objective functions at both levels are not necessarily convex. As a result, the lower-level solution set $\mathcal{X}_g^*$ could be a nonconvex set, making it intractable to achieve any form of global optimality. Consequently, in nonconvex simple bilevel optimization, similar to its single-level counterpart, the primary objective is to find a near-stationary point rather than a near-optimal solution, as defined in [2, 12, 16].

The search for near-stationary points in nonconvex simple bilevel problems has been addressed by only a few works. Among these, Gong et al. [3] proposed the Dynamic Barrier Gradient Descent (DBGD) algorithm, which employs a dynamic barrier constraint on the search direction at each iteration. By adaptively balancing objectives $f$ and $g$ with a dynamic combination coefficient, it guides the optimization trajectory. It was originally introduced to solve the constrained problem:

$$\min_{\mathbf{x} \in \mathbb{R}^n} \ f(\mathbf{x}) \quad \text{s.t.} \quad g(\mathbf{x}) \leq c, \tag{2}$$

where $f$ and $g$ are smooth but possibly nonconvex, $c \geq g^*$, and $g^*$ is the minimum value of $g$. Note that the analysis in [3] is limited to the continuous-time limit behavior of DBGD (step size $\eta \to 0$). Specifically, it was shown that the continuous-time dynamics of DBGD converge at a rate of $\mathcal{O}(1/t)$ in terms of the violation of the Karush–Kuhn–Tucker (KKT) conditions of Problem (2) with the assumption of bounded dual iterates, i.e., $\max_t \lambda_t < +\infty$. For the specific choice of $c = g^*$, the problem in (2) becomes equivalent to the simple bilevel problem in (1). However, in this case, the assumption of bounded dual iterates is violated, rendering the associated theoretical guarantees inapplicable. Under the additional assumption that $\|\nabla f\|$ and $\|\nabla g\|$ are uniformly bounded, the presented continuous-time convergence rate deteriorates to $\mathcal{O}(\max(1/t^{2/\tau}, 1/t^{1-1/\tau}))$ for any user-defined $\tau > 1$. More importantly, their analysis *does not* hold when considering the discrete time case (step size $\eta > 0$).

Another closely related work is [5], which introduced BLOOP (BiLevel Optimization with Orthogonal Projection) for stochastic nonconvex simple bilevel problems. The core idea of BLOOP is to project the upper-level gradient onto the space orthogonal to the lower-level gradient. However, their analysis is limited to a non-asymptotic convergence rate of $\mathcal{O}(1/K^{1/4})$ for the lower-level objective, where $K$ is the number of iterations, without providing any rate guarantees for the upper-level objective.

Motivated by the above discussion, we aim to address the following research question:

*Is it possible to design a first-order method with discrete-time guarantees for both levels of the nonconvex simple bilevel problem in* (1) *under the given assumptions?*

**Contributions.** Motivated by this research question, We begin by defining a first-order stationarity metric for nonconvex simple bilevel problems in Section 3, which intuitively identifies points where no significantly better solution exists in a local neighborhood. In Section 3.1, we relate this notion to existing stationarity concepts in the literature. We then develop and analyze a practical variant of the dynamic barrier gradient descent (DBGD) method proposed in [3], providing theoretical guarantees for its convergence in discrete time. The specifics of our main contributions are as follows:

(i) We define an $(\epsilon_f, \epsilon_g)$-stationary point for nonconvex simple bilevel optimization as a point $\hat{\mathbf{x}}$ for which there exists $\lambda \geq 0$ such that $\|\nabla f(\hat{\mathbf{x}}) + \lambda \nabla g(\hat{\mathbf{x}})\|^2 \leq \epsilon_f$ and $\|\nabla g(\hat{\mathbf{x}})\|^2 \leq \epsilon_g$, where $\epsilon_f$ and $\epsilon_g$ specify the desired stationarity accuracy for the upper and lower levels. We also discuss how this notion relates to existing stationarity metrics in the constrained and bilevel optimization literature.

(ii) We show that to achieve an $(\epsilon_f, \epsilon_g)$-stationary point of the considered nonconvex simple bilevel problem, the studied method has a complexity of $\mathcal{O}(\max(\epsilon_f^{-\frac{3+p}{1+p}}, \epsilon_g^{-\frac{3+p}{2}}))$, where $p \geq 0$. This is the

first explicit (discrete-time) complexity bound that guarantees stationarity at both levels for nonconvex simple bilevel problems.

Further connections between first-order methods for convex and nonconvex simple bilevel problems and the DBGD framework are discussed in Appendix C.

## 1.1 Related Work

**Convex simple bilevel problems.** Most studies on the "simple bilevel optimization problem" focus on cases where both the upper-level and lower-level objective functions are convex. Various approaches have been developed to address such convex simple bilevel optimization problems, including regularization-based methods [7–10, 17], penalty-based methods [18], sequential averaging methods [19, 20], online-learning-based methods [21], lower-level linearized based methods [2, 4, 12], and bisection-based methods [22, 16]. However, due to the convexity assumption underlying these algorithms, they are not applicable in our setting.

**Nonconvex constrained minimization problems.** The simple bilevel problem (1) can also be reformulated as Problem (2) with $c = g^*$. This reformulation suggests that existing method for functionally constrained optimization [23–30] could be applied to solve (2). However, this approach presents several challenges. First, since $g$ is nonconvex, estimating $g^*$ to high accuracy is intractable. Moreover, Problem (2) does not satisfy strict feasibility, and most common constraint qualifications, such as the Mangasarian-Fromovitz Constraint Qualification, do not hold. Hence, many results, such as those in [27, 30, 29] may no longer hold.

**Nonconvex general bilevel problems.** Recent works on nonconvex general bilevel optimization [31–36] rely on different stationarity notions or assumptions and are thus not directly applicable to our setting. For instance, the works in [31, 32] define stationarity based on the norm of the hyper-gradient, which may be ill-defined in the simple bilevel setting where no upper-level variables are present, rendering it an invalid metric. Moreover, most existing approaches assume the Polyak–Łojasiewicz (PL) condition [33, 35, 31, 32] for the lower-level problem—an assumption not made in our setting, thereby invalidating the convergence guarantees established in those works. A detailed comparison between algorithms for general and simple bilevel problems, along with a discussion of different stationarity metrics and other related methods, is provided in Appendix D.

After our submission, we became aware of a concurrent work [37], which provides a similar analysis for a variant of our algorithm. Their study further demonstrates the algorithm's effectiveness in the context of machine unlearning, revealing an additional and complementary application area to our theoretical and empirical findings.

## 2 Assumptions

We denote $g^* > -\infty$ and $f^* > -\infty$ as the minimum value of $g$ and $f$, respectively. We next formally state our assumptions.

**Assumption 2.1.** *We assume these conditions hold:*

(i) *$f$ has bounded gradients, i.e., $\|\nabla f(\mathbf{x})\| \leq G_f < \infty$ for any $\mathbf{x} \in \mathbb{R}^n$.*

(ii) *$f$ is continuously differentiable, and $\nabla f$ is $L_f$-Lipschitz, i.e., $\|\nabla f(\mathbf{x}) - \nabla f(\mathbf{y})\| \leq L_f \|\mathbf{x} - \mathbf{y}\|$.*

(iii) *$g$ is continuously differentiable, and $\nabla g$ is $L_g$-Lipschitz, i.e., $\|\nabla g(\mathbf{x}) - \nabla g(\mathbf{y})\| \leq L_g \|\mathbf{x} - \mathbf{y}\|$.*

## 3 Stationarity Metric

This section introduces our performance metric for evaluating convergence rates of algorithms for Problem (1). While objective value gap is commonly used in convex settings [2, 10, 12, 22, 16], it is intractable here due to the potential nonconvexity of $f$ and $g$. Instead, we measure stationarity, defining an approximate stationary point as a point $\hat{\mathbf{x}} \in \mathbb{R}^n$ where no significantly better solution exists nearby—i.e., one that lowers $g$, or lowers $f$ without increasing $g$. We next present first-order conditions that capture this notion, followed by their interpretation.

**Definition 3.1.** *Given $\epsilon_f \geq 0$ and $\epsilon_g \geq 0$, a point $\hat{\mathbf{x}} \in \mathbb{R}^n$ is an $(\epsilon_f, \epsilon_g)$-stationary point of Problem* (1) *if there exists a scalar $\lambda \geq 0$ such that:*

$$\|\nabla g(\hat{\mathbf{x}})\|^2 \leq \epsilon_g \quad and \quad \|\nabla f(\hat{\mathbf{x}}) + \lambda \nabla g(\hat{\mathbf{x}})\|^2 \leq \epsilon_f. \tag{3}$$

Definition 3.1 consists of two conditions that measure the stationarity of a given solution $\hat{\mathbf{x}}$. The first condition requires that the gradient of the lower-level objective $g$ at $\hat{\mathbf{x}}$ is small, meaning that $\mathbf{x}$ is near stationary for $g$. To better interpret the second condition, we first introduce a decomposition that expresses the upper-level gradient $\nabla f(\hat{\mathbf{x}})$ as the sum of two orthogonal components: one parallel to $\nabla g(\hat{\mathbf{x}})$ and the other orthogonal to it. Specifically, we write $\nabla f(\hat{\mathbf{x}}) = \nabla f_{\|}(\hat{\mathbf{x}}) + \nabla f_{\perp}(\hat{\mathbf{x}})$, where $\nabla f_{\|}(\hat{\mathbf{x}})$ is the component parallel to $\nabla g(\hat{\mathbf{x}})$ and $\nabla f_{\perp}(\hat{\mathbf{x}})$ is the component orthogonal to $\nabla g(\hat{\mathbf{x}})$. Using this decomposition, we can further express:

$$\nabla f(\hat{\mathbf{x}}) + \lambda \nabla g(\hat{\mathbf{x}}) = \nabla f_{\perp}(\hat{\mathbf{x}}) + (\nabla f_{\|}(\hat{\mathbf{x}}) + \lambda \nabla g(\hat{\mathbf{x}})), \tag{4}$$

where the first term is orthogonal to $\nabla g(\hat{\mathbf{x}})$ and the second part (in parenthesis) is parallel to it. Hence, the conditions in (3) ensure that all the following norms are small: (i) $\|\nabla g(\hat{\mathbf{x}})\|$, (ii) $\|\nabla f_{\perp}(\hat{\mathbf{x}})\|$, and (iii) $\|\nabla f_{\|}(\hat{\mathbf{x}}) + \lambda \nabla g(\hat{\mathbf{x}})\|$. Interestingly, as we shall show in the convergence analysis, these terms do not necessarily diminish at the same rate.

This decomposition offers key insights into the final output. If the first two terms are small, the resulting point satisfies two key conditions: (i) the gradient norm of the lower-level problem is small, and (ii) the component of the upper-level gradient orthogonal to the lower-level gradient is also small. In other words, in directions that do not negatively impact the lower-level objective, there is no remaining energy to further decrease the upper-level objective.

The third term being small leads to two possible cases. If $\nabla f_{\|}(\hat{\mathbf{x}})$ is aligned with $\nabla g(\hat{\mathbf{x}})$, then since $\|\nabla g(\hat{\mathbf{x}})\|$ is small and $\lambda > 0$, it follows that $\nabla f_{\|}(\hat{\mathbf{x}})$ is also small. Combined with the smallness of $\|\nabla f_{\perp}(\hat{\mathbf{x}})\|$, this implies $\|\nabla f(\hat{\mathbf{x}})\|$ is small as well. On the other hand, if $\nabla f_{\|}(\hat{\mathbf{x}})$ is in the opposite direction of $\nabla g(\hat{\mathbf{x}})$, we cannot make a definitive statement about $\|\nabla f_{\|}(\hat{\mathbf{x}})\|$, as $\lambda$ could be large. Hence, any point satisfying the conditions in (3) must fall into one of the following two cases:

- **Case I:** At $\hat{\mathbf{x}}$, both $\|\nabla g(\hat{\mathbf{x}})\|$ and $\|\nabla f(\hat{\mathbf{x}})\|$ are small, indicating near-stationarity for both the lower- and upper-level objectives.

- **Case II:** At $\hat{\mathbf{x}}$, $\|\nabla g(\hat{\mathbf{x}})\|$ is small and the gradient of $f$ has minimal energy in directions orthogonal to $\nabla g(\hat{\mathbf{x}})$ and its remaining energy (norm) is in the opposite direction of $\nabla g(\hat{\mathbf{x}})$.

In both cases, we reach a point where further decreasing $f$ would necessarily increase $g$, indicating that no additional progress can be made without violating the constraint. This means the objective function cannot be significantly improved in its local neighborhood without incurring greater infeasibility. This concept is formally characterized in the following lemma.

**Lemma 3.1.** *A point $\hat{\mathbf{x}} \in \mathbb{R}^n$ is an $(\epsilon_f, \epsilon_g)$-stationary point of Problem* (1) *if and only if the following holds: for any $\delta > 0$, there exists a radius $\hat{r} > 0$ such that for all $0 < r < \hat{r}$:*

- *For any $\mathbf{x}$ satisfying $\|\mathbf{x} - \hat{\mathbf{x}}\| \leq r$, we have $g(\mathbf{x}) \geq g(\hat{\mathbf{x}}) - (1 + \delta)\sqrt{\epsilon_g}\|\hat{\mathbf{x}} - \mathbf{x}\|$.*

- *For any $\mathbf{x}$ satisfying $\|\mathbf{x} - \hat{\mathbf{x}}\| \leq r$ and $g(\mathbf{x}) \leq g(\hat{\mathbf{x}})$, we have $f(\mathbf{x}) \geq f(\hat{\mathbf{x}}) - (1 + \delta)\sqrt{\epsilon_f}\|\hat{\mathbf{x}} - \mathbf{x}\|$.*

The first condition of the lemma guarantees that the lower-level objective $g$ cannot be improved by more than $\mathcal{O}(\sqrt{\epsilon_g}\|\hat{\mathbf{x}} - \mathbf{x}\|)$ locally. The second condition further shows that the upper-level objective cannot be significantly improved without negatively impacting $g$.

## 3.1 Connections with Other Stationarity Metrics

In this section, we examine the connection between our proposed stationarity metrics in Definition 3.1 and existing notions of stationarity in both constrained optimization and bilevel optimization literature.

### 3.1.1 Connection with the Metrics in Constrained Optimization Literature

We note that Definition 3.1 is closely related to approximate KKT conditions for a reformulation of Problem (1). Specifically, recall $g^* = \min_{\mathbf{x} \in \mathbb{R}^n} g(\mathbf{x})$ and the constraint in (1) is equivalent to

$g(\mathbf{x}) - g^* \leq 0$. Thus, Problem (1) can be reformulated as Problem (2) with $c = g^*$. Given a point $\mathbf{x}$ and its Lagrange multiplier $\lambda \geq 0$, the KKT conditions for (2) are:

$$g(\mathbf{x}) - g^* \leq 0, \quad \lambda \geq 0, \quad \nabla f(\mathbf{x}) + \lambda \nabla g(\mathbf{x}) = 0, \quad \lambda(g(\mathbf{x}) - g^*) = 0.$$

However, since Problem (2) with $c = g^*$ is not strictly feasible, Slater's condition does not hold, and the KKT conditions may not hold at an optimal solution. Moreover, since $g$ is nonconvex, enforcing strict feasibility is intractable. To resolve this, the literature on nonconvex constrained optimization has considered relaxed stationarity conditions such as the *scaled KKT conditions* [23–27]. When specialized to Problem (1), these papers aim to find a point $\hat{\mathbf{x}}$ that satisfies *one of* the following for given accuracy parameters $\epsilon_p$ and $\epsilon_d$: (i) $\hat{\mathbf{x}}$ satisfies an approximate scaled KKT conditions, i.e.,

$$g(\hat{\mathbf{x}}) - g^* \leq \epsilon_p, \ \lambda \geq 0, \ \|\nabla f(\hat{\mathbf{x}}) + \lambda \nabla g(\hat{\mathbf{x}})\| \leq \epsilon_d(1 + \lambda). \tag{5}$$

Here, the accuracy of the last condition is proportional to the Lagrange multiplier $\lambda$. (ii) $\hat{\mathbf{x}}$ is an infeasible stationary point of the constraint function, i.e.,

$$g(\hat{\mathbf{x}}) - g^* \geq 0.99\epsilon_p, \quad \|\nabla g(\hat{\mathbf{x}})\| \leq \epsilon_d. \tag{6}$$

Moreover, [28] considered the stronger *unscaled KKT conditions*, where (5) is replaced by

$$g(\hat{\mathbf{x}}) - g^* \leq \epsilon_p, \ \lambda \geq 0, \ \|\nabla f(\hat{\mathbf{x}}) + \lambda \nabla g(\hat{\mathbf{x}})\| \leq \epsilon_d. \tag{7}$$

Unlike the scaled KKT conditions, the accuracy requirement in the last condition does not depend on the multiplier $\lambda$. We observe that our Definition 3.1 implies the unscaled KKT conditions. Suppose $\hat{\mathbf{x}}$ is an approximate $(\epsilon_d^2, \epsilon_d^2)$-stationary point. Then, if $g(\hat{\mathbf{x}}) - g^* \geq 0.99\epsilon_p$, the condition in (6) holds; otherwise, the condition in (7) is satisfied.

### 3.1.2 Connection with the Metrics in Bilevel Optimization Literature

In this section, we also relate our proposed stationarity metric for the original simple bilevel problem (1) to those used in common reformulations in the bilevel optimization literature. A widely used reformulation is the value-function-based approach, defined as follows:

$$\min_{\mathbf{x} \in \mathbb{R}^n} f(\mathbf{x}) \quad \text{s.t.} \quad g(\mathbf{x}) - g^* = 0 \tag{8}$$

However, the KKT conditions of the value-function reformulation are not necessary for optimality, as standard constraint qualifications may be violated—even when the lower-level objective satisfies additional conditions such as the Polyak-Łojasiewicz (PL) condition [33, 35]. Instead, we aim to connect our proposed stationarity condition with the KKT conditions of the gradient-based reformulation. Before establishing this connection, we introduce the following definition and assumption.

**Definition 3.2** (($\epsilon_p, \epsilon_d$)-KKT conditions [35]). *A gradient-based reformulation of Problem* (1) *is*

$$\min_{x \in \mathbb{R}^n} f(\mathbf{x}), \quad s.t. \quad \nabla g(\mathbf{x}) = 0, \tag{9}$$

*A point $\hat{\mathbf{x}}$ is an $(\epsilon_p, \epsilon_d)$-KKT point of Problem* (9) *if there exists $\mathbf{w} \in \mathbb{R}^n$ such that*

$$\|\nabla f(\hat{\mathbf{x}}) + \nabla^2 g(\hat{\mathbf{x}})\mathbf{w}\|^2 \leq \epsilon_p, \quad \|\nabla g(\hat{\mathbf{x}})\|^2 \leq \epsilon_d \tag{10}$$

Note that the reformulation (9) is equivalent to Problem (1) when $g$ satisfies the PL condition. To analyze the relationship between Definition 3.1 and 3.2, we introduce the following assumption.

**Assumption 3.1** (Local Error Bound [38]). *There exists $c > 0$ such that for $\epsilon$ small enough and for any $\mathbf{x}$ satisfying $\|\nabla g(\mathbf{x})\| \leq \epsilon$, we have $\textbf{dist}(\mathbf{x}, \nabla g^{-1}(\{0\})) \leq c\|\nabla g(\mathbf{x})\|$.*

This local error bound condition is implied by a local PL inequality, which itself is a relaxation of the global PL condition. We are now ready to connect our proposed stationarity metric with the $(\epsilon_p, \epsilon_d)$-KKT conditions of the gradient-based reformulated problem. The Proof is in Appendix A.2.

**Theorem 3.2.** *Suppose Assumption 3.1 holds and $\nabla^2 g(\mathbf{x})$ is $L_H$-Lipschitz. If a point $\hat{\mathbf{x}} \in \mathbb{R}^n$ is an $(\epsilon_f, \epsilon_g)$-stationary point of Problem* (1) *for some $\epsilon_f, \epsilon_g > 0$, then it is an $(\epsilon_p, \epsilon_d)$-KKT point of Problem* (9) *for $\epsilon_p = \mathcal{O}(\epsilon_f + \lambda\|\nabla g(\hat{\mathbf{x}})\|\epsilon_g)$ and $\epsilon_d = \epsilon_g$.*

Theorem 3.2 implies that although the KKT solutions of Problem (9) typically rely on second-order information of the lower-level objective, they can still be approximated using first-order methods. In particular, this result holds without requiring any constraint qualification (CQ) conditions commonly assumed in the bilevel optimization literature [3, 33, 39].

# 4 Algorithmic Framework

To efficiently find a stationary point for the nonconvex simple bilevel problem in (1), we adopt the dynamic barrier gradient descent (DBGD) framework in [3]. It was first proposed for the constrained optimization problem in (2), with theoretical guarantees established only in the continuous-time limit. One of our contributions is applying this framework to the nonconvex simple bilevel problem (1) and establishing the first discrete-time stationarity guarantees. The core idea of DBGD is to choose a descent direction that aligns with the upper-level gradient while minimizing its impact on the lower-level problem. Specifically, consider the general update rule

$$\mathbf{x}_{k+1} = \mathbf{x}_k - \eta_k \mathbf{d}_k, \tag{11}$$

where $\eta_k > 0$ is a step size and $\mathbf{d}_k$ is a descent direction. For the simple bilevel problem of interest, we seek a vector $\mathbf{d}_k$ that balances progress on both objectives. When the lower-level objective is far from optimal, the focus is on minimizing it while ensuring that any reduction in the upper-level objective $f$ does not hinder the decrease of $g$. As the lower-level objective nears optimality, priority shifts to minimizing $f$, which may require a controlled increase in $g$ to keep iterates $\mathbf{x}$ within or close to the solution set $\mathcal{X}_g^*$. It turns out that both properties can be achieved if $\mathbf{d}_k$ is selected as

$$\mathbf{d}_k = \operatorname*{argmin}_{\mathbf{d} \in \mathbb{R}^n} \|\nabla f(\mathbf{x}_k) - \mathbf{d}\|^2 \qquad \text{s.t.} \quad \nabla g(\mathbf{x}_k)^\top \mathbf{d} \geq \phi(\mathbf{x}_k). \tag{12}$$

Here, $\phi : \mathbb{R}^d \to \mathbb{R}^+$ is a non-negative function that controls the inner product between the selected direction and the gradient of the lower-level problem. Specifically, Gong et al. [3] propose the choice $\phi(\mathbf{x}) = \min\{\alpha(g(\mathbf{x}_k) - g^*), \beta\|\nabla g(\mathbf{x}_k)\|^2\}$. This represents one possible design, and as elaborated in Appendix C, alternative choices for $\phi$ give rise to other methods studied in the literature. The main property of $\phi$ is that it should capture some form of infeasibility for the original problem, i.e., suboptimality in the lower-level problem.

A key point is that $\mathbf{d}_k$ is chosen as the closest vector to the upper-level gradient $\nabla f$ while maintaining a positive angle with the lower-level gradient. The set of feasible directions depends on how far the current point is from feasibility. If $g(\mathbf{x}_k) = g^*$, i.e., $\phi(\mathbf{x}_k) = 0$, any direction with an angle less than 90 degrees is feasible, allowing us to reduce $f$ without increasing $g$ (up to first-order). But if $\phi(\mathbf{x}_k)$ is large, we prioritize reducing $g$ by choosing a direction closely aligned with $\nabla g$.

**Close form solution of the subproblem.** Since (12) is a quadratic convex program with a single inequality constraint, its optimal solution can be explicitly expressed as

$$\mathbf{d}_k = \nabla f(\mathbf{x}_k) + \lambda_k \nabla g(\mathbf{x}_k), \tag{13}$$

where $\lambda_k$ can be computed as follows:

$$\lambda_k = \max\left\{ \frac{\phi(\mathbf{x}_k) - \nabla f(\mathbf{x}_k)^\top \nabla g(\mathbf{x}_k)}{\|\nabla g(\mathbf{x}_k)\|^2}, 0 \right\} \tag{14}$$

Hence, our method of interest with stepsize $\eta_k$ can be easily implemented by following the update

$$\mathbf{x}_{k+1} = \mathbf{x}_k - \eta_k (\nabla f(\mathbf{x}_k) + \lambda_k \nabla g(\mathbf{x}_k)). \tag{15}$$

**Our choice of the subproblem.** To establish convergence guarantees for the nonconvex simple bilevel problem, we analyze the version of the discussed algorithm that incorporates $\phi(\mathbf{x}_k) = \beta_k \|\nabla g(\mathbf{x}_k)\|^2$ in its update. This choice is motivated by the fact that, in nonconvex lower-level problems, the gradient norm is the most computationally tractable measure of suboptimality. In this case, the expression for the parameter $\lambda_k$ introduced in (14) can be simplified as

$$\lambda_k = \max\left\{ \frac{\beta_k \|\nabla g(\mathbf{x}_k)\|^2 - \nabla f(\mathbf{x}_k)^\top \nabla g(\mathbf{x}_k)}{\|\nabla g(\mathbf{x}_k)\|^2}, 0 \right\} \tag{16}$$

In Section 5, we establish the convergence rate of the update that follows the update in (15) when $\lambda_k$ is computed based on the expression in (16).

# 5 Convergence Analysis

In this section, we analyze the convergence rate of a variant of the dynamic barrier descent method, which follows the updates in (15) and (16) to solve the nonconvex simple bilevel problem in (1). As

discussed, our goal is to find a point $\hat{x}$ that satisfies the conditions in (3). To achieve this, it suffices to show that, for at least one iterate of the method, both the lower-level gradient norm $\|\nabla g(\mathbf{x}_k)\|$ and the update direction norm $\|\mathbf{d}_k\|$ are small. Given that $\mathbf{d}_k = \nabla f(\mathbf{x}_k) + \lambda_k \nabla g(\mathbf{x}_k)$ and that $\lambda_k$ in our algorithm is always non-negative, this guarantees the desired convergence in Definition 3.1.

Our starting point is to use the smoothness property of the objective functions $f$ and $g$ (Assumption 2.1(ii) and (iii)) to derive a descent-type lemma. This leads to an upper bound on $\|\mathbf{d}_k\|$ and $\|\nabla g(\mathbf{x}_k)\|$ in each iteration, which is shown in the following lemma. The proof is in Appendix B.1.

**Lemma 5.1.** *Suppose Assumption 2.1 holds and let $\{\mathbf{x}_k\}$ be the iterates generated by* (15) *and* (16) *with a constant step size $\eta_k \equiv \eta$ and a constant hyperparameter $0 \leq \beta_k \equiv \beta \leq 1$. Define $\Delta f_k = f(\mathbf{x}_k) - f(\mathbf{x}_{k+1})$ and $\Delta g_k = g(\mathbf{x}_k) - g(\mathbf{x}_{k+1})$. We have*

$$(1 - \frac{\eta L_f}{2})\|\mathbf{d}_k\|^2 \leq \frac{\Delta f_k}{\eta} + \lambda_k \beta \|\nabla g(\mathbf{x}_k)\|^2, \tag{17}$$

$$\beta \|\nabla g(\mathbf{x}_k)\|^2 \leq \frac{\Delta g_k}{\eta} + \frac{L_g}{2}\eta \|\mathbf{d}_k\|^2. \tag{18}$$

Lemma 5.1 shows that $\|\mathbf{d}_k\|$ can be upper bounded in terms of $\Delta f = f(\mathbf{x}_k) - f(\mathbf{x}_{k+1})$ and $\|\nabla g(\mathbf{x}_k)\|$, while $\|\nabla g(\mathbf{x}_k)\|$ can, in turn, be upper bounded in terms of $\Delta g = g(\mathbf{x}_k) - g(\mathbf{x}_{k+1})$ and $\|\mathbf{d}_k\|$. A natural strategy, therefore, is to combine the two inequalities (17) and (18) and construct a potential function of the form $\|\mathbf{d}_k\|^2 + c\|\nabla g(\mathbf{x}_k)\|^2$, where $c$ is an appropriate constant. This would be easy to achieve if $\lambda_k$ is uniformly bounded by an absolute constant $M$. Indeed, in this case, by adding (17) and (18) multiplied by $2M$, and further assuming that $\eta \leq \frac{1}{L_f + 2ML_g}$, we obtain $\frac{1}{2}\|\mathbf{d}_k\|^2 + M\|\nabla g(\mathbf{x}_k)\|^2 \leq \frac{\Delta f_k + 2M\Delta g}{\eta}$. Applying the standard telescoping argument then yields a convergence rate of $\mathcal{O}(\frac{1}{\eta K})$ for both $\|\mathbf{d}_k\|^2$ and $\|\nabla g(\mathbf{x}_k)\|^2$.

However, a key challenge is that we *do not* have an a priori upper bound on $\lambda_k$, which prevents us from setting a constant step size $\eta$ that depends on such a bound. Moreover, such a uniform upper bound on $\lambda_k$ may not even exist at all, as $\lambda_k$ could diverge to infinity when $\mathbf{x}_k$ approaches a near-stationary point of the lower-level objective $g$. To see this, recall the expression for $\lambda_k$ in (16) and the decomposition of the upper-level gradient $\nabla f(\mathbf{x}_k) = \nabla f_\|(\mathbf{x}_k) + \nabla f_\perp(\mathbf{x}_k)$, where $\nabla f_\|(\mathbf{x}_k)$ is the component parallel to $\nabla g(\mathbf{x}_k)$ and $\nabla f_\perp(\mathbf{x}_k)$ is the component orthogonal to $\nabla g(\mathbf{x}_k)$. When $\nabla f_\|(\mathbf{x}_k)$ is in the opposite direction of $\nabla g(\mathbf{x}_k)$, we have $\lambda_k = \beta - \frac{\nabla f(\mathbf{x}_k)^\top \nabla g(\mathbf{x}_k)}{\|\nabla g(\mathbf{x}_k)\|^2} = \beta + \frac{\|\nabla f_\|(\mathbf{x}_k)\|\|\nabla g(\mathbf{x}_k)\|}{\|\nabla g(\mathbf{x}_k)\|^2} = \beta + \frac{\|\nabla f_\|(\mathbf{x}_k)\|}{\|\nabla g(\mathbf{x}_k)\|}$. Thus, as $\|\nabla g(\mathbf{x}_k)\|$ approaches zero, $\lambda_k$ diverges whenever $\nabla f_\|(\mathbf{x}_k)$ is nonzero—that is, when the upper-level gradient retains nonzero energy in the opposite direction of $\nabla g(\mathbf{x}_k)$.

The following lemma presents our first attempt to use the boundedness of the upper-level gradient (Assumption 2.1(i)) to control the magnitude of $\lambda_k$. The proof is in Appendix B.2.

**Lemma 5.2.** *Recall the expression of $\lambda_k$ in* (16)*. If Assumption 2.1 holds, then $\lambda_k \leq \beta + \frac{G_f}{\|\nabla g(\mathbf{x}_k)\|}$.*

*Remark* 5.1. This result shows $\lambda_k$ is proportional to $\frac{1}{\|\nabla g(\mathbf{x}_k)\|}$ rather than being uniformly bounded by a constant. As a result, for the $\lambda$ generated by DBGD, we have $\epsilon_p = \mathcal{O}(\epsilon_f + \epsilon_g)$ in Theorem 3.2.

However, this bound still suffers from the same issue: it is vacuous as $\|\nabla g(\mathbf{x}_k)\|$ approaches zero.

To address this challenge, we construct a new potential function that circumvents the need to explicitly upper-bound $\lambda_k$ by a constant. The key observation is that $\lambda_k$ appears in (17) only as a coefficient of the term $\|\nabla g(\mathbf{x}_k)\|^2$. Hence, while $\lambda_k$ is potentially unbounded, the total contribution of the term $\lambda_k \|\nabla g(\mathbf{x}_k)\|^2$ in (17) can be controlled by $\beta \|\nabla g(\mathbf{x}_k)\|^2 + G_f \|\nabla g(\mathbf{x}_k)\|$, which converges to zero as long as $\|\nabla g(\mathbf{x}_k)\|$ diminishes. This suggests that we may still obtain a meaningful bound on $\|\mathbf{d}_k\|$ by appropriately combining (17) and (18). This is stated in the next lemma.

**Lemma 5.3.** *Consider the updates in* (15)*. If Assumptions 2.1 hold, $\eta \leq \frac{1}{L_f + L_g}$, and $\beta \leq 1$, then*

$$\|\mathbf{d}_k\|^2 \leq \frac{2(\Delta f_k + \beta \Delta g_k)}{\eta} + 2\sqrt{\beta}G_f \sqrt{\frac{\Delta g_k}{\eta} + \frac{L_g}{2}\eta \|\mathbf{d}_k\|^2}. \tag{19}$$

Note that this is an *implicit* inequality for $\|\mathbf{d}_k\|$, as $\|\mathbf{d}_k\|$ is also present in the right-hand side under the square root. To obtain an explicit bound, we first present the following intermediate lemma.

**Lemma 5.4.** *Suppose $x \geq 0$ and $x \leq A + B\sqrt{x}$, where $A \in \mathbb{R}$ and $B \geq 0$. Then $x \leq 2A + B^2$.*

To apply Lemma 5.4 and obtain an explicit upper bound on $\|\mathbf{d}_k\|^2$, we first manipulate (19) to match the form of the inequality in Lemma 5.4. Specifically, adding $\frac{2\Delta g_k}{L_g \eta^2}$ to both sides of (19) and defining $S_k \triangleq \|\mathbf{d}_k\|^2 + \frac{2\Delta g_k}{L_g \eta^2}$, we obtain the following inequality

$$S_k \leq \frac{2(\Delta f_k + \beta \Delta g_k)}{\eta} + \frac{2\Delta g_k}{L_g \eta^2} + \sqrt{\beta} G_f \sqrt{2L_g \eta} \sqrt{S_k}. \tag{20}$$

Applying Lemma 5.4 to (20) yields $S_k \leq \frac{4(\Delta f_k + \beta \Delta g_k)}{\eta} + \frac{4\Delta g_k}{L_g \eta^2} + 2\beta G_f^2 L_g \eta$. Since $S_k = \|\mathbf{d}_k\|^2 + \frac{2\Delta g_k}{L_g \eta^2}$, it follows

$$\|\mathbf{d}_k\|^2 \leq \frac{4(\Delta f_k + \beta \Delta g_k)}{\eta} + \frac{2\Delta g_k}{L_g \eta^2} + 2\beta G_f^2 L_g \eta, \tag{21}$$

which provides an upper bound on $\|\mathbf{d}_k\|^2$. As we shall see later, this inequality together with (18) will be the key to constructing our new potential function. We can now proceed to our main theorem, which characterizes the convergence rate of the algorithm. The proof is in Appendix B.5.

**Theorem 5.5.** *Suppose Assumption 2.1 holds and let $\{\mathbf{x}_k\}$ be generated by (15) and (16) with a constant step size $\eta_k \equiv \eta = \frac{1}{LK^{1/(3+p)}}$, where $L := L_f + L_g$, and hyperparameter $\beta_k \equiv \beta = (L\eta)^p = \frac{1}{K^{p/(3+p)}}$, where $p \geq 0$. Further, define $\Delta_f := f(\mathbf{x}_0) - \inf f$, and $\Delta_g := g(\mathbf{x}_0) - g^*$. Then, there exists an index $k^* \in \{1, \cdots, K\}$ such that*

$$\|\nabla g(\mathbf{x}_{k^*})\|^2 \leq \frac{4L\Delta_f}{K^{3/(3+p)}} + \frac{4L\Delta_g}{K} + \frac{3L\Delta_g + 2G_f^2}{K^{2/(3+p)}} \tag{22}$$

$$\|\nabla f(\mathbf{x}_{k^*}) + \lambda_{k^*} \nabla g(\mathbf{x}_{k^*})\|^2 \leq \frac{8L\Delta_f}{K^{(2+p)/(3+p)}} + \frac{8L\Delta_g}{K^{(2+2p)/(3+p)}} + \frac{6L^2\Delta_g + 4G_f^2}{L_g K^{(1+p)/(3+p)}} \tag{23}$$

As a corollary, the algorithm based on the updates in (15) and (16) finds an $(\epsilon_f, \epsilon_g)$-stationary point after $\mathcal{O}(\max(\epsilon_f^{-\frac{3+p}{1+p}}, \epsilon_g^{-\frac{3+p}{2}}))$ iterations, where $p \geq 0$. If we want to balance the rates of the upper and lower levels, we can choose $p = 1$, i.e. $\beta = \mathcal{O}(\eta)$, in which case the algorithm finds an $(\epsilon_f, \epsilon_g)$-stationary point after $\mathcal{O}(\max(\epsilon_f^{-2}, \epsilon_g^{-2}))$ iterations. To our knowledge, this is the first discrete-time non-asymptotic guarantee to the stationary points for nonconvex simple bilevel optimization.

*Remark* 5.2. Gong et al. [3] analyzed the continuous-time limit of the algorithm in (15) and (16). However, their continuous-time analysis does not account for the additional error introduced by approximating functions $f$ and $g$ by their first-order Taylor expansions. As a result, their convergence result does not directly translate into a concrete convergence bound for the discrete-time algorithm. Some of our key contributions include addressing the additional discretization error—requiring the solution of an implicit inequality (cf. Lemma 5.3) and careful selection of the step size $\eta$ and hyperparameter $\beta$—as well as removing the common assumption of uniformly bounded $\|\nabla g\|$.

*Remark* 5.3. Note that the sequence $\{\lambda_k\}_{k \geq 0}$ may go to infinity asymptotically, which can be a potential issue for Definition 3.1. Specifically, as $K \to \infty$, the algorithm may converge to a point $\mathbf{x}_\infty$ where $\nabla g(\mathbf{x}_\infty) = 0$, in which case there may not be a finite $\lambda$ such that $x_\infty$ and $\lambda$ satisfy the second condition in Definition 3.1. However, in this limiting case, the above issue can be addressed by considering the alternative stationarity condition in Definition 3.2. In particular, the second condition in Definition 3.1 is replaced by $\|\nabla f(\hat{\mathbf{x}}) + \nabla^2 g(\hat{\mathbf{x}})\mathbf{w}\|^2 \leq \epsilon_p$ for some bounded vector $\mathbf{w}$. Note that our algorithm ensures that $\lambda_k = \mathcal{O}(1/\|\nabla g(\mathbf{x}_k)\|)$, so the product $\lambda_k \|\nabla g(\mathbf{x}_k)\|$ remains bounded and has a finite limit point. By applying Theorem 3.2, we can show that the limit point of our algorithm satisfies Definition 3.2 with $\epsilon_p = O(\epsilon_f + \epsilon_g)$ under Assumption 3.1. Finally, we note that reaching a point with exactly vanishing gradient is rare in practice, and since $\lambda_k = \mathcal{O}(1/\|\nabla g(\mathbf{x}_k)\|)$, the sequence $\lambda_k$ remains finite in all practical cases.

## 6 Numerical Experiments

While the primary focus of this work is theoretical, we include a set of numerical experiments to illustrate the behavior of the proposed algorithm and to support our theoretical findings. Since

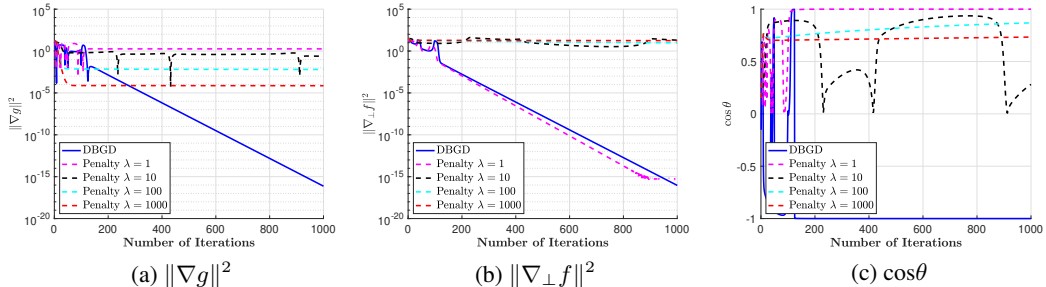

| (a) $\|\nabla g\|^2$ | (b) $\|\nabla_\perp f\|^2$ | (c) $\cos\theta$ |

Figure 1: The performance of DBGD compared with Penalty methods with different choices of $\lambda$ on Problem (24) in terms of $\|\nabla g\|^2$, $\|\nabla_\perp f\|^2$, and $\cos\theta$.

prior studies [3, 5] have already demonstrated the strong empirical performance of DBGD in large-scale neural network training tasks, we do not repeat such experiments here. Instead, we evaluate DBGD on deterministic optimization problems, which align more closely with the scope of this paper. For comparison, we consider a penalty-based method with updates of the form $\mathbf{x}_{k+1} = \mathbf{x}_k - \eta_k(\nabla f(\mathbf{x}_k) + \lambda \nabla g(\mathbf{x}_k))$, where $\lambda \geq 0$ is fixed. We justify the use of this penalty method as a baseline in Appendix D, and provide full experimental details in Appendix E.

**Toy Example.** We study the following nonconvex simple bilevel problem,

$$\min_{\mathbf{x} \in \mathbb{R}^2} (x_1 + \frac{\pi}{20})^2 + (x_2 + 1)^2 \text{ s.t. } \mathbf{x} \in \operatorname*{argmin}_{\mathbf{z} \in \mathbb{R}^2}(z_2 - \sin(10z_1))^2 \qquad (24)$$

Based on Definition 3.1, we use $\|\nabla g\|$, $\|\nabla_\perp f\|$, and $\cos\theta$—where $\theta$ is the angle between $\nabla g$ and $\nabla f$—to measure stationarity. Specifically, $\|\nabla g\|$ corresponds to the first condition in Definition 3.1, while $\|\nabla_\perp f\|$ and $\cos\theta$ reflect the second condition. As shown in Figure 1, DBGD outperforms the penalty methods across all the metrics, regardless of the choice of penalty parameter $\lambda$. In Figure 1 (a), although increasing the penalty parameter $\lambda$ in the penalty method accelerates early-stage convergence, the lower-level stationarity metric $\|\nabla g\|$ ultimately plateaus. In Figure 1 (b), only small values of $\lambda$ effectively reduce the norm of the orthogonal component of $\nabla f$. In Figure 1 (c), the penalty methods produce iterates where the angle between $\nabla f$ and $\nabla g$ remains less than $90°$, indicating that the gradients are not fully conflicting and that further improvement is possible. In contrast, DBGD consistently improves both $\|\nabla g\|^2$ and $\|\nabla_\perp f\|^2$ in Figure 1 (a) and (b). Moreover, the angle between $\nabla g$ and $\nabla f$ approaches $180°$ in Figure 1 (c), indicating that further local improvement is not possible. Taken together, these observations show that the iterate generated by DBGD satisfies Definition 3.1.

**Matrix Factorization.** We formulate matrix factorization [40–42] as a simple bilevel problem that seeks to approximate a symmetric matrix via $\mathbf{M} \approx \mathbf{U}\mathbf{U}^\top$, where $\mathbf{U}$ is a low-rank tall matrix, while simultaneously optimizing a secondary criterion.

$$\min_{\mathbf{U} \in \mathbb{R}^{n \times r}} f(\mathbf{U}) \quad \text{s.t.} \quad \mathbf{U} \in \operatorname*{argmin}_{\mathbf{V} \in \mathbb{R}^{n \times r}} g(\mathbf{V}) = \|\mathbf{M} - \mathbf{V}\mathbf{V}^\top\|_F^2 \qquad (25)$$

In our experiments, the lower-level objective is the reconstruction loss, while the upper-level objective $f(\mathbf{U})$ is designed to promote sparsity. Since the $\ell_1$-norm is non-smooth, one can adopt a smooth approximation such as $f_1(\mathbf{U}) = \sum_{i,j} \sqrt{U_{ij}^2 + \alpha}$. Alternatively, a log-smooth sparsity penalty can be used [43]: $f_2(\mathbf{U}) = \sum_{i,j} \log(1 + U_{ij}^2/\alpha)$. Both $f_1$ and $f_2$ are smooth and encourage sparsity in $\mathbf{U}$.

Figure 2 presents the results of applying DBGD and the penalty method with various choices of $\beta$ or $\lambda$ to solve Problem (25). Similar to the previous experiment, we use $\|\nabla g\|$ and $\|\nabla_\perp f\|$ as convergence metrics, corresponding to the two conditions in Definition 3.1. Additionally, we report the objective values of both the upper- and lower-level problems, which represent the sparsity and reconstruction loss, respectively. As shown in Figures 2 (a) and (c), the solutions obtained by DBGD consistently outperform those produced by the penalty method with respect to both stationarity metrics, $\|\nabla g\|$ and $\|\nabla_\perp f\|$. This superiority holds across a wide range of hyperparameter values, regardless of the choice of $\beta$ for DBGD or $\lambda$ for the penalty method, highlighting the effectiveness of DBGD in achieving stationarity. In addition to the stationarity metrics, DBGD also consistently achieves low

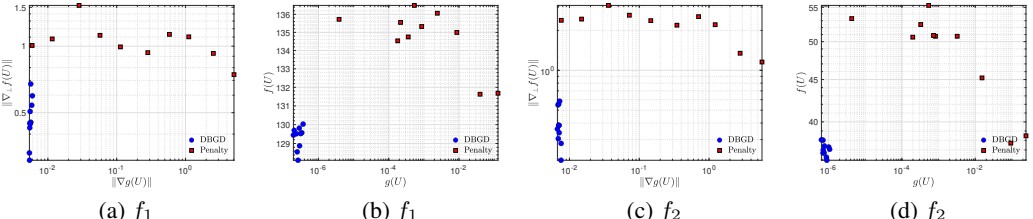

|     |     |     |     |
|-----|-----|-----|-----|
| (a) $f_1$ | (b) $f_1$ | (c) $f_2$ | (d) $f_2$ |

Figure 2: The performance of DBGD compared with Penalty methods on Problem (25) in terms of $\|\nabla g\|^2$, $\|\nabla_\perp f\|^2$, $g$, and $f$. Blue dots indicate the performance of DBGD with different choices of $\beta$, while red dots show the performance of the penalty method with varying penalty parameters $\lambda$.

reconstruction loss $g(\mathbf{U})$ and sparsity penalty $f(\mathbf{U})$ across a wide range of $\beta$ values. In contrast, the performance of the penalty-based methods is highly sensitive to the choice of the penalty parameter $\lambda$, often resulting in suboptimal trade-offs between reconstruction and sparsity. These differences are clearly illustrated in Figures 2 (b) and (d), further demonstrating the effectiveness of DBGD.

# 7 Conclusion

In this paper, we focused on nonconvex simple bilevel problems and introduced the definition of $(\epsilon_f, \epsilon_g)$-stationary points as a stationarity metric for this problem class, examining its relationship with existing metrics in the literature. We then established a novel non-asymptotic analysis for a variant of the dynamic barrier gradient descent algorithm framework from [3], demonstrating a convergence rate of $\mathcal{O}(\max(\epsilon_f^{-\frac{3+p}{1+p}}, \epsilon_g^{-\frac{3+p}{2}}))$, where $p \geq 0$, for achieving $(\epsilon_f, \epsilon_g)$-stationary points for nonconvex simple bilevel problems.

# Acknowledgements

The research of J. Cao, R. Jiang and A. Mokhtari is supported in part by NSF Grants 2127697, 2019844, and 2112471, ARO Grant W911NF2110226, the Machine Learning Lab (MLL) at UT Austin, and the Wireless Networking and Communications Group (WNCG) Industrial Affiliates Program. The research of E. Yazdandoost Hamedani is supported by NSF Grant 2127696.

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

## Appendix / supplemental material

## A    Omitted Proofs in Section 3

### A.1    Proof of Lemma 3.1

First, we show that if the point $\hat{\mathbf{x}}$ is an $(\epsilon_f, \epsilon_g)$-stationary point as defined in Definition 3.1, then the two conditions in Lemma 3.1 are satisfied. For any $\delta > 0$, let $\hat{r} = \min\{2\delta\sqrt{\epsilon_g}/L_g, 2\delta\sqrt{\epsilon_f}/(\lambda L_g + L_f)\}$. For any $\mathbf{x}$ satisfying $\|\mathbf{x} - \hat{\mathbf{x}}\| \leq r \leq \hat{r}$, Using the fact that $g$ is $L_g$-smooth and $\|\nabla g(\hat{\mathbf{x}})\|^2 \leq \epsilon_g$, it holds that

$$g(\mathbf{x}) \geq g(\hat{\mathbf{x}}) + \langle \nabla g(\hat{\mathbf{x}}), \mathbf{x} - \hat{\mathbf{x}} \rangle - \frac{L_g}{2}\|\hat{\mathbf{x}} - \mathbf{x}\|^2$$
$$\geq g(\hat{\mathbf{x}}) - \sqrt{\epsilon_g}\|\mathbf{x} - \hat{\mathbf{x}}\| - \delta\sqrt{\epsilon_g}\|\hat{\mathbf{x}} - \mathbf{x}\| = g(\hat{\mathbf{x}}) - (1+\delta)\sqrt{\epsilon_g}\|\mathbf{x} - \hat{\mathbf{x}}\|,$$

where we used $\|\mathbf{x} - \hat{\mathbf{x}}\| \leq r \leq 2\delta\sqrt{\epsilon_g}/L_g$ in the second inequality. Thus, the first condition in Lemma 3.1 is satisfied. Moreover, Consider any $\mathbf{x}$ that satisfies $\|\mathbf{x} - \hat{\mathbf{x}}\| \leq r$ and $g(\mathbf{x}) \leq g(\hat{\mathbf{x}})$. Since $f$ is $L_f$-smooth, it holds that $f(\mathbf{x}) \geq f(\hat{\mathbf{x}}) + \langle \nabla f(\hat{\mathbf{x}}), \mathbf{x} - \hat{\mathbf{x}} \rangle - \frac{L_f}{2}\|\hat{\mathbf{x}} - \mathbf{x}\|^2$. By using $\|\nabla f(\hat{\mathbf{x}}) + \lambda \nabla g(\hat{\mathbf{x}})\| \leq \sqrt{\epsilon_f}$, we further have

$$f(\mathbf{x}) \geq f(\hat{\mathbf{x}}) + \langle \nabla f(\hat{\mathbf{x}}) + \lambda \nabla g(\hat{\mathbf{x}}), \mathbf{x} - \hat{\mathbf{x}} \rangle - \lambda \langle \nabla g(\hat{\mathbf{x}}), \mathbf{x} - \hat{\mathbf{x}} \rangle - \frac{L_f}{2}\|\hat{\mathbf{x}} - \mathbf{x}\|^2$$
$$\geq f(\hat{\mathbf{x}}) - \sqrt{\epsilon_f}\|\mathbf{x} - \hat{\mathbf{x}}\| - \lambda \langle \nabla g(\hat{\mathbf{x}}), \mathbf{x} - \hat{\mathbf{x}} \rangle - \frac{L_f}{2}\|\hat{\mathbf{x}} - \mathbf{x}\|^2.$$

Using the smoothness of $g$, we also have $g(\mathbf{x}) \geq g(\hat{\mathbf{x}}) + \langle \nabla g(\hat{\mathbf{x}}), \mathbf{x} - \hat{\mathbf{x}} \rangle - \frac{L_g}{2}\|\mathbf{x} - \hat{\mathbf{x}}\|^2$. Hence, we get $-\langle \nabla g(\hat{\mathbf{x}}), \mathbf{x} - \hat{\mathbf{x}} \rangle \geq -\frac{L_g}{2}\|\mathbf{x} - \hat{\mathbf{x}}\|^2$. Thus, this leads to

$$f(\mathbf{x}) \geq f(\hat{\mathbf{x}}) - \sqrt{\epsilon_f}\|\mathbf{x} - \hat{\mathbf{x}}\| - \lambda\frac{L_g}{2}\|\hat{\mathbf{x}} - \mathbf{x}\|^2 - \frac{L_f}{2}\|\hat{\mathbf{x}} - \mathbf{x}\|^2.$$

Since $\|\hat{\mathbf{x}} - \mathbf{x}\| \leq r \leq 2\delta\sqrt{\epsilon_f}/(\lambda L_g + L_f)$, we obtain $f(\mathbf{x}) \geq f(\hat{\mathbf{x}}) - (1+\delta)\sqrt{\epsilon_f}\|\mathbf{x} - \hat{\mathbf{x}}\|$. This shows that the second condition in Lemma 3.1 is also satisfied.

For the other direction, assume that $\hat{\mathbf{x}}$ satisfies both conditions in Lemma 3.1. Consider any direction $\mathbf{d} \in \mathbb{R}^n$. Then Condition (i) implies that, for all $t$ small enough, we have $g(\hat{\mathbf{x}} - t\mathbf{d}) \geq g(\hat{\mathbf{x}}) - (1+\delta)\epsilon_g t\|\mathbf{d}\|$, which can be rewritten as $\frac{g(\hat{\mathbf{x}}) - g(\hat{\mathbf{x}} - t\mathbf{d})}{t} \leq (1+\delta)\epsilon_g\|\mathbf{d}\|$. By taking the limit $t \to 0$, we obtain $\langle \nabla g(\hat{\mathbf{x}}), \mathbf{d} \rangle \leq (1+\delta)\epsilon_g\|\mathbf{d}\|$. By taking $\mathbf{d} = \nabla g(\hat{\mathbf{x}})$, this implies that $\|\nabla g(\hat{\mathbf{x}})\| \leq (1+\delta)\epsilon_g$. Since this holds for any $\delta > 0$, taking the limit $\delta \to 0$ yields $\|\nabla g(\hat{\mathbf{x}})\| \leq \epsilon_g$. Moreover, let $\mathbf{d} \in \mathbb{R}^n$ be any direction that satisfies $\langle \nabla g(\hat{\mathbf{x}}), \mathbf{d} \rangle > 0$. Then for all $t$ small enough, it holds that $g(\hat{\mathbf{x}} - t\mathbf{d}) \leq g(\hat{\mathbf{x}})$. Thus, using Condition (ii), we have

$$f(\hat{\mathbf{x}} - t\mathbf{d}) \geq f(\hat{\mathbf{x}}) - (1+\delta)\epsilon_f t\|\mathbf{d}\| \quad \Rightarrow \quad \frac{f(\hat{\mathbf{x}}) - f(\hat{\mathbf{x}} - t\mathbf{d})}{t} \leq (1+\delta)\epsilon_f\|\mathbf{d}\|.$$

Similarly, by taking the limits $t \to 0$ and $\delta \to 0$, we obtain $\langle \nabla f(\hat{\mathbf{x}}), \mathbf{d} \rangle \leq \epsilon_f\|\mathbf{d}\|$. Since this holds for any $\mathbf{d}$ that satisfies $\langle \nabla g(\hat{\mathbf{x}}), \mathbf{d} \rangle > 0$, continuity ensures that it also holds for any $\mathbf{d}$ such that $\langle \nabla g(\hat{\mathbf{x}}), \mathbf{d} \rangle \geq 0$. If $\langle \nabla f(\mathbf{x}), \nabla g(\mathbf{x}) \rangle \geq 0$, then by setting $\mathbf{d} = \nabla f(\mathbf{x})$, we obtain that $\|\nabla f(\hat{\mathbf{x}})\| \leq \epsilon_f$. Otherwise, if $\langle \nabla f(\mathbf{x}), \nabla g(\mathbf{x}) \rangle < 0$, let $\lambda = -\frac{\langle \nabla f(\hat{\mathbf{x}}), \nabla g(\hat{\mathbf{x}}) \rangle}{\|\nabla g(\hat{\mathbf{x}})\|^2} > 0$ and set $\mathbf{d} = \nabla f(\hat{\mathbf{x}}) + \lambda \nabla g(\hat{\mathbf{x}})$. Note that this choice of $\lambda$ ensures that $\langle \nabla g(\hat{\mathbf{x}}), \mathbf{d} \rangle = 0$, and hence $\langle \nabla f(\hat{\mathbf{x}}), \mathbf{d} \rangle = \|\mathbf{d}\|^2 \leq \epsilon_f\|\mathbf{d}\|$, which implies that $\|\mathbf{d}\| \leq \epsilon_f$. This completes the proof.

### A.2    Proof of Theorem 3.2

Suppose $\hat{\mathbf{x}}$ is an $(\epsilon_f, \epsilon_g)$-stationary point of Problem (1), the second inequality in Definition 3.2 is satisfied with $\epsilon_d = \epsilon_g$. Now, we start to prove the first inequality in Definition 3.2 by setting $\mathbf{w} = \lambda(\hat{\mathbf{x}} - \mathbf{x}^*)$, where $\mathbf{x}^*$ denotes the stationary point closest to $\hat{\mathbf{x}}$.

$$\|\nabla f(\hat{\mathbf{x}}) + \nabla^2 g(\hat{\mathbf{x}})\mathbf{w}\| \leq \|\nabla f(\hat{\mathbf{x}}) + \nabla^2 g(\mathbf{x}^*)\mathbf{w}\| + \|\nabla^2 g(\hat{\mathbf{x}}) - \nabla^2 g(\mathbf{x}^*)\|\|\mathbf{w}\|$$

$$\leq \|\nabla f(\hat{\mathbf{x}}) + \lambda\nabla g(\hat{\mathbf{x}})\| + \lambda\|\nabla g(\hat{\mathbf{x}}) - \nabla^2 g(\mathbf{x}^*)(\hat{\mathbf{x}} - \mathbf{x}^*)\| + \lambda L_H\|\hat{\mathbf{x}} - \mathbf{x}^*\|^2$$

$$\leq \epsilon_f + \lambda\|\nabla g(\hat{\mathbf{x}}) - \nabla g(\mathbf{x}^*) - \nabla^2 g(\mathbf{x}^*)(\hat{\mathbf{x}} - \mathbf{x}^*)\| + \lambda L_H\|\hat{\mathbf{x}} - \mathbf{x}^*\|^2$$

$$\leq \epsilon_f + 2\lambda L_H\|\hat{\mathbf{x}} - \mathbf{x}^*\|^2 \leq \epsilon_f + \lambda\|\nabla g(\hat{\mathbf{x}})\| \cdot 2L_H c^2\epsilon_g$$

$$= \mathcal{O}(\epsilon_f + \lambda\|\nabla g(\hat{\mathbf{x}})\|\epsilon_g)$$

where the second and fourth inequalities follow from the Lipschitz continuity of $\nabla^2 g(\mathbf{x})$, the third follows from the second condition in Definition 3.1, and the last follows from Assumption 3.1. Hence, the first condition in Definition 3.2 holds with $\epsilon_p = \mathcal{O}(\epsilon_f + \lambda\|\nabla g(\hat{\mathbf{x}})\|\epsilon_g)$.

# B  Omitted Proofs in Section 5

## B.1  Proof of Lemma 5.1

From Assumption 2.1, $f$ has an $L_f$-Lipschitz continuous gradient, hence,

$$f(\mathbf{x}_{k+1}) - f(\mathbf{x}_k) \leq -\eta\nabla f(\mathbf{x}_k)^\top\mathbf{d}_k + \frac{L_f}{2}\eta^2\|\mathbf{d}_k\|^2$$

$$= -\eta(\nabla f(\mathbf{x}_k) - \mathbf{d}_k)^\top\mathbf{d}_k - \eta(1 - \frac{L_f}{2}\eta)\|\mathbf{d}_k\|^2$$

$$= \eta\lambda_k\nabla g(\mathbf{x}_k)^\top\mathbf{d}_k - \eta(1 - \frac{L_f}{2}\eta)\|\mathbf{d}_k\|^2$$

where in the last equality we used $\nabla f(\mathbf{x}_k) = \mathbf{d}_k - \lambda_k\nabla g(\mathbf{x}_k)$. Since $\mathbf{d}_k$ is the optimal solution of subproblem (12) with the corresponding optimal dual multiplier $\lambda_k$, the complementarity slackness implies that $\lambda_k(\nabla g(\mathbf{x}_k)^\top\mathbf{d}_k - \beta\|\nabla g(\mathbf{x}_k)\|^2) = 0$. Hence, we further obtain

$$f(\mathbf{x}_{k+1}) - f(\mathbf{x}_k) \leq -\eta(1 - \frac{L_f}{2}\eta)\|\mathbf{d}_k\|^2 + \eta\lambda_k\beta\|\nabla g(\mathbf{x}_k)\|^2.$$

By dividing both sides by $\eta$ and rearranging the inequality, we obtain (17).

Moreover, from Assumption 2.1, $g$ has an $L_g$-Lipschitz continuous gradient, which implies that

$$g(\mathbf{x}_{k+1}) - g(\mathbf{x}_k) \leq -\eta\nabla g(\mathbf{x}_k)^\top\mathbf{d}_k + \frac{L_g}{2}\eta^2\|\mathbf{d}_k\|^2 \leq -\eta\beta\|\nabla g(\mathbf{x}_k)\|^2 + \frac{L_g}{2}\eta^2\|\mathbf{d}_k\|^2,$$

where we used $\nabla g(\mathbf{x}_k)^\top\mathbf{d}_k \geq \beta\|\nabla g(\mathbf{x}_k)\|^2$ from (12) in the last inequality. Dividing both sides by $\eta$ and rearranging the inequality yields (18).

## B.2  Proof of Lemma 5.2

By Assumption 2.1, the gradient of $f$ is bounded by $G_f$. Thus, we have

$$\lambda_k \leq \beta + \frac{|\langle\nabla f(\mathbf{x}_k), \nabla g(\mathbf{x}_k)\rangle|}{\|\nabla g(\mathbf{x}_k)\|^2} \leq \beta + \frac{G_f}{\|\nabla g(\mathbf{x}_k)\|}.$$

This completes the proof.

## B.3  Proof of Lemma 5.3

By combining Lemma 5.2 with (17), we have $(1 - \frac{\eta L_f}{2})\|\mathbf{d}_k\|^2 \leq \frac{\Delta f_k}{\eta} + \beta^2\|\nabla g(\mathbf{x}_k)\|^2 + \beta G_f\|\nabla g(\mathbf{x}_k)\|$. Substituting the upper bound on $\|\nabla g(\mathbf{x}_k)\|$ in (18) and combining terms, we arrive at $(1 - \frac{\eta(L_f + \beta L_g)}{2})\|\mathbf{d}_k\|^2 \leq \frac{\Delta f_k + \beta\Delta g_k}{\eta} + \sqrt{\beta}G_f\sqrt{\frac{\Delta g_k}{\eta} + \frac{L_g}{2}\eta\|\mathbf{d}_k\|^2}$. Since $\eta \leq \frac{1}{L_f + L_g} \leq \frac{1}{L_f + \beta L_g}$, the left side of this inequality can be lower bounded by $\frac{1}{2}\|\mathbf{d}_k\|^2$. By multiplying both sides by 2 the claim follows.

### B.4 Proof of Lemma 5.4

Since $x \leq A + B\sqrt{x}$, we have $(\sqrt{x} - \frac{B}{2})^2 \leq A + \frac{B^2}{4}$, which further implies $\sqrt{x} - \frac{B}{2} \leq \sqrt{A + \frac{B^2}{4}}$. By adding $\frac{B}{2}$ to both sides, taking the square, and using Young's inequality we obtain $x \leq (\sqrt{A + \frac{B^2}{4}} + \frac{B}{2})^2 = A + \frac{B^2}{2} + B\sqrt{A + \frac{B^2}{4}} \leq A + \frac{B^2}{2} + \frac{B^2}{4} + (A + \frac{B^2}{4}) = 2A + B^2$. This completes the proof.

### B.5 Proof of Theorem 5.5

Multiplying (18) by $\frac{1}{L_g \eta}$ and adding it to (21), implies

$$\frac{\|\mathbf{d}_k\|^2}{2} + \frac{\beta \|\nabla g(\mathbf{x}_k)\|^2}{L_g \eta} \leq \frac{4(\Delta f_k + \beta \Delta g_k)}{\eta} + \frac{3\Delta g_k}{L_g \eta^2} + 2\beta G_f^2 L_g \eta.$$

Define the potential function as $\mathcal{G}_k \triangleq \frac{1}{2}\|\mathbf{d}_k\|^2 + \frac{\beta}{L_g \eta}\|\nabla g(\mathbf{x}_k)\|^2$. Averaging the above inequality over $k = 0$ to $K - 1$ and noting that $\sum_{k=0}^{K-1} \Delta f_k = f(\mathbf{x}_0) - f(\mathbf{x}_K) \leq f(\mathbf{x}_0) - \inf f = \Delta_f$ and $\sum_{k=0}^{K-1} \Delta g_k = g(\mathbf{x}_0) - g(\mathbf{x}_K) \leq g(\mathbf{x}_0) - g^* = \Delta_g$, we obtain

$$\frac{1}{K} \sum_{k=0}^{K-1} \mathcal{G}_k \leq \frac{4(\Delta_f + \beta \Delta_g)}{\eta K} + \frac{3\Delta_g}{L_g \eta^2 K} + 2\beta G_f^2 L_g \eta.$$

Since $\eta = \frac{1}{LK^{1/(3+p)}}$ and $\beta = \frac{1}{K^{p/(3+p)}}$, by letting $k^* = \operatorname{argmin}_{0 \leq k \leq K-1} \mathcal{G}_k$, we get

$$\mathcal{G}_{k^*} \leq \frac{4L\Delta_f}{K^{(2+p)/(3+p)}} + \frac{4L\Delta_g}{K^{(2+2p)/(3+p)}} + \frac{3L^2 \Delta_g}{L_g K^{(1+p)/(3+p)}} + \frac{2G_f^2}{K^{(1+p)/(3+p)}}.$$

Finally, since $\mathcal{G}_{k^*} = \frac{1}{2}\|\mathbf{d}_{k^*}\|^2 + \frac{\beta}{L_g \eta}\|\nabla g(\mathbf{x}_{k^*})\|^2$, it follows that $\|\mathbf{d}_{k^*}\|^2 \leq 2\mathcal{G}_{k^*}$ and $\|\nabla g(\mathbf{x}_{k^*})\|^2 \leq \frac{L_g \eta}{\beta}\mathcal{G}_{k^*} = \frac{L_g \mathcal{G}_{k^*}}{LK^{(1-p)/(3+p)}}$. By the definition $\mathbf{d}_k = \nabla f(\mathbf{x}_k) + \lambda_k \nabla g(\mathbf{x}_k)$ and the fact that $\lambda_k \geq 0$, the proof is complete.

## C Other Choices of $\phi(\mathbf{x})$ and their connection to methods considered in the literature.

In this section, we briefly discuss the connection between other methods studied in the literature and the general algorithmic framework described in (11)-(12).

**Lower-level linearization based methods.** If we set $\phi(\mathbf{x}) = \alpha(g(\mathbf{x}) - g^*)$ in the update (12), where $\alpha = 1/\eta$, the resulting method closely aligns with the lower-level linearization-based approach introduced in [2]. This method was originally developed to solve simple bilevel optimization problems with a *convex* lower-level objective. The key idea of this type of method is to construct a halfspace to approximate the lower-level solution set $\mathcal{X}_g^*$. Specifically, the approximated set is constructed using a linear approximation of the lower-level objective as follows,

$$\mathcal{X}_k = \{\mathbf{x} \in \mathbb{R}^n : g(\mathbf{x}_k) + \nabla g(\mathbf{x}_k)^\top (\mathbf{x} - \mathbf{x}_k) \leq g^*\}$$

If $g$ is convex, then the constructed set $\mathcal{X}_k$ contains $\mathcal{X}_g^*$ for all $k$. The update of the projection variant of the algorithm in [2] is as follows,

$$\mathbf{x}_{k+1} = \Pi_{\mathcal{X}_k}(\mathbf{x}_k - \eta \nabla f(\mathbf{x}_k))$$

which would be equivalent to

$$\mathbf{x}_{k+1} = \operatorname*{argmin}_{\mathbf{x}} \|\mathbf{x} - (\mathbf{x}_k - \eta \nabla f(\mathbf{x}_k))\|^2 \quad \text{s.t.} \quad g(\mathbf{x}_k) + \nabla g(\mathbf{x}_k)^\top (\mathbf{x} - \mathbf{x}_k) \leq g^*$$

Through change of variables and defining $\mathbf{d} = (\mathbf{x}_k - \mathbf{x})/\eta$, we can equivalently reformulate the above subproblem as

$$\mathbf{d}_k = \operatorname*{argmin}_{\mathbf{d}} \|\mathbf{d} - \nabla f(\mathbf{x}_k)\|^2 \quad \text{s.t.} \quad \nabla g(\mathbf{x}_k)^\top \mathbf{d} \geq (g(\mathbf{x}_k) - g^*)/\eta.$$

This is a special instance of (12) with $\phi(\mathbf{x}) = (g(\mathbf{x}) - g^*)/\eta$. This choice of $\phi(\mathbf{x})$ is suitable for convex problems, as the solution set $\mathcal{X}_g^*$ is convex and can be contained within $\mathcal{X}_k$. However, when the lower-level loss is nonconvex, $\mathcal{X}_g^*$ is also nonconvex, meaning the inclusion $\mathcal{X}_g^* \subseteq \mathcal{X}_k$ is not guaranteed. To address this, $\phi(\mathbf{x})$ must be adapted, and using the gradient norm offers a natural extension to the nonconvex case.

**Orthogonal projection methods**. BiLevel Optimization with Orthogonal Projection (BLOOP) [5] was recently proposed for stochastic simple bilevel optimization. Its key idea is projecting the upper-level gradient to be orthogonal to the lower-level gradient. In the deterministic version, the descent direction $\mathbf{d}_k$ for the update $\mathbf{x}_{k+1} = \mathbf{x}_k - \eta_k \mathbf{d}_k$ is chosen as

$$\mathbf{d}_k = \beta \nabla g(\mathbf{x}_k) + \left[ \nabla f(\mathbf{x}_k) - \frac{\nabla f(\mathbf{x}_k)^\top \nabla g(\mathbf{x}_k)}{\|\nabla g(\mathbf{x}_k)\|^2} \nabla g(\mathbf{x}_k) \right]$$

The second part of $\mathbf{d}_k$ is the projection of the upper-level gradient onto the orthogonal space of the lower-level gradient. If we rearrange the terms in $\mathbf{d}_k$, $\mathbf{d}_k$ is equivalent to

$$\mathbf{d}_k = \underset{\mathbf{d}}{\arg\min} \ \|\mathbf{d} - \nabla f(\mathbf{x}_k)\|^2 \qquad \text{s.t.} \quad \nabla g(\mathbf{x}_k)^\top \mathbf{d} = \beta \|\nabla g(\mathbf{x}_k)\|^2.$$

This is a special case of (12) with $\phi(\mathbf{x}) = \beta \|\nabla g(\mathbf{x})\|^2$, but with an equality constraint instead of an inequality. Solving the equality-constrained subproblem with the chosen $\phi(\mathbf{x})$ ensures convergence of the lower-level objective but not the upper-level one [5]. In contrast, we show that solving the inequality-constrained problem also guarantees convergence for the upper level.

# D   Connections with Algorithms for General Bilevel Problems

In this section, we discuss why most algorithms designed for general bilevel problems are not directly applicable to our simple bilevel setting and highlight the connections between the two classes of algorithms. In the general form of bilevel problems, the upper-level function $f$ may also depend on an additional variable $\mathbf{y} \in \mathbb{R}^m$ that in turn influences the lower-level problem:

$$\min_{\mathbf{x} \in \mathbb{R}^n, \mathbf{y} \in \mathbb{R}^m} f(\mathbf{x}, \mathbf{y}) \quad \text{s.t.} \quad \mathbf{x} \in \arg\min_{\mathbf{z} \in \mathbb{R}^n} g(\mathbf{z}, \mathbf{y})$$

However, in our considered simple bilevel setting, there is no additional upper-level variable. As a result, the upper-level updates present in algorithms for general bilevel problems become invalid. When these updates are removed, some algorithms—such as those in [44, 45, 35]—reduce to standard gradient descent on $g$, i.e., $\mathbf{x}_{k+1} = \mathbf{x}_k - \eta_k \nabla g(\mathbf{x}_k)$. Many other methods [33, 34, 31, 46, 32] reduce to the update,

$$\mathbf{x}_{k+1} = \mathbf{x}_k - \eta_k \left( \nabla f(\mathbf{x}_k) + \lambda_k \nabla g(\mathbf{x}_k) \right),$$

which we refer to as the penalty method for nonconvex simple bilevel problems. We include this method as a baseline in our experiments in Section 6. The key challenge for the penalty method lies in selecting an appropriate penalty parameter $\lambda_k$. The choices of $\lambda_k$ used in general bilevel problems are not suitable for the simple bilevel setting, as they are based on different stationarity metrics. Therefore, determining the appropriate value of $\lambda_k$ for this method requires a tailored analysis specific to the simple bilevel setting. Note that DBGD algorithm essentially provides a dynamic scheme for selecting $\lambda_k$, as described in (14).

## D.1   Connections with Stationarity Metrics for General Bilevel Problems

Besides the algorithms themselves, the stationarity metrics for general bilevel problems are also not directly applicable to the simple bilevel setting. For instance, [47, 31, 32] adopt the norm of the hyper-gradient as a measure of stationarity. Recall that the hyper-objective [48] is defined as follows :

$$\min_{\mathbf{y} \in \mathbb{R}^m} \varphi(\mathbf{y}), \quad \text{where } \varphi(\mathbf{y}) = \min_{\mathbf{x} \in X^*(\mathbf{y})} f(\mathbf{x}, \mathbf{y}),$$

where $X^*(\mathbf{y}) \triangleq \arg\min_{\mathbf{z}} g(\mathbf{z}, \mathbf{y})$. However, in the simple bilevel setting without upper-level variables $\mathbf{y}$, the norm of the hyper-gradient constant and thus fails to serve as a valid metric. Furthermore, most existing approaches rely on strong convexity or the Polyak–Łojasiewicz (PL) condition for the lower-level problem—assumptions that are violated in our case, where the hyper-gradient may not even be well-defined.

Other works, such as [35], consider alternative stationarity metrics. When rewritten in the context of our simple bilevel setting, their condition becomes: there exists $\mathbf{w} \in \mathbb{R}^n$ such that

$$\|\nabla^2 g(\hat{\mathbf{x}})(\nabla f(\hat{\mathbf{x}}) + \nabla^2 g(\hat{\mathbf{x}})\mathbf{w})\|^2 \leq \epsilon_f, \quad \|\nabla g(\hat{\mathbf{x}})\|^2 \leq \epsilon_g.$$

Intuitively, the first condition ensures that the component of $\nabla f(\hat{\mathbf{x}}) + \nabla^2 g(\hat{\mathbf{x}})w$ projected onto the kernel of $\nabla^2 g(\hat{\mathbf{x}})$ is small, i.e.,

$$\text{Proj}_{\text{Ker}(\nabla^2 g(\hat{\mathbf{x}}))} \left( \nabla f(\hat{\mathbf{x}}) + \nabla^2 g(\hat{\mathbf{x}})\mathbf{w} \right) \approx 0.$$

This stationary metric is generally weaker than the metric defined in Definition 3.2.

### D.2 Additional Related Works on General Bilevel Problems

To go beyond strongly convex lower-level objectives, additional assumptions on the lower-level problem are necessary to ensure meaningful guarantees, particularly in light of the negative results for general bilevel optimization with merely convex lower-level objectives [32]. A common strategy is to assume that the nonconvex lower-level objective satisfies the Polyak–Łojasiewicz (PL) condition. Specifically, a penalty-based gradient method was introduced in [34] for both unconstrained and constrained nonconvex-PL bilevel optimization. Later, [35] proposed GALET, a Hessian-vector-product-based method with non-asymptotic convergence guarantees to the modified KKT points of a gradient-based reformulation. In [31], nonconvex bilevel optimization under the proximal error-bound (EB) condition was studied, which is analogous to the PL condition. More recently, in [36], a Hessian/Jacobian-free method was developed that achieves optimal convergence complexity for nonconvex-PL bilevel problems. Besides imposing the PL condition on the lower-level problem, these works also rely on different additional assumptions. For example, [33] additionally assumes that both the upper- and lower-level function values, as well as the norms of their gradients, are bounded, and the lower-level optimal solution is unique. The work in [35] requires both PL and convexity assumptions on the lower-level problem to guarantee convergence. The studies in [31] and [32] impose the condition that a weighted sum of the upper- and lower-level objectives satisfies the PL condition. Finally, in [36] it is assumed that $\nabla^2 g(\mathbf{x})$ is non-singular at the minimizer of $g$.

### D.3 On the Role of the PL Condition

The PL condition plays a central role in the analyses of the aforementioned works in general bilevel optimization. For example, [32] heavily relies on the fact that the PL condition induces a "strongly convex subspace" around any minimizer of the lower-level objective. This structural property enables the adaptation of proof techniques similar to those in [46], which developed an algorithm for general bilevel problems with a strongly convex lower-level objective. Essentially, in general bilevel settings, the PL condition ensures the continuity of the hyper-objective $\varphi(\mathbf{y})$, thereby guaranteeing the existence of the hyper-gradient. This facilitates rapid convergence to a neighborhood of $X^*(\mathbf{y})$. However, in our considered simple bilevel setting, the hyper-objective and its gradient are not well-defined, and we instead rely on alternative stationarity metrics. Consequently, the PL condition is less applicable and offers limited benefit compared to its role in general bilevel problems.

## E Experiments Details

In this section, we include more details of the numerical experiments in Section 6. All simulations are implemented using MATLAB R2022a on a PC running macOS Sonoma with an Apple M1 Pro chip and 16GB Memory.

**Toy Example.** Recall that for Problem (24), the optimal solution set of the lower-level problem is given by $\mathcal{X}_g^* = \{\mathbf{x} \in \mathbb{R}^2 : x_2 = \sin(10x_1)\}$. The optimal solution of the bilevel problem is $\mathbf{x}^* = \left(-\frac{\pi}{20}, -1\right)$. We apply DBGD using $\phi(\mathbf{x}) = \|\nabla g(\mathbf{x}_k)\|^2$, i.e., with $\beta = 1$, and also employ the Penalty methods introduced in Section D with $\lambda \in \{1, 10, 100, 1000\}$. Both methods are initialized at the point $\mathbf{x}_0 = (-3, -1)$, using a base stepsize of $\eta = 10^{-2}$ and a total of $K = 10^3$ iterations. Since the penalty methods become unstable for large values of $\lambda$, we further scale the stepsize by a factor of $1/(1 + \lambda)$ in each independent run.

**Matrix Factorization.** For Problem (25), we set $n = r = 10$ to generate $\mathbf{U}_*$ and construct $\mathbf{M} = \mathbf{U}_* \mathbf{U}_*^\top + \epsilon \mathbf{I}_n$, where $\epsilon \sim \mathcal{N}(0, 0.01)$ and $\mathbf{I}_n \in \mathbb{R}^{n \times n}$ denotes the identity matrix. We

apply DBGD with $\beta \in \{0.1, 0.2, 0.3, 0.4, 0.5, 0.6, 0.7, 0.8, 0.9, 1\}$ and compare it against the penalty methods described in Section D, using $\lambda \in \{1, 2, 5, 10, 20, 50, 100, 200, 500, 1000\}$. Both methods use a stepsize of $\eta = 10^{-5}$ and are run for $K = 10^6$ iterations. Since the penalty methods become unstable for large values of $\lambda$, we further scale the stepsize by a factor of $1/(1+\lambda)$ in each independent run. The hyperparameter $\alpha$ in both $f_1$ and $f_2$ is set to 1.

## F    Additional Experiments

### F.1    Different Stationary Points

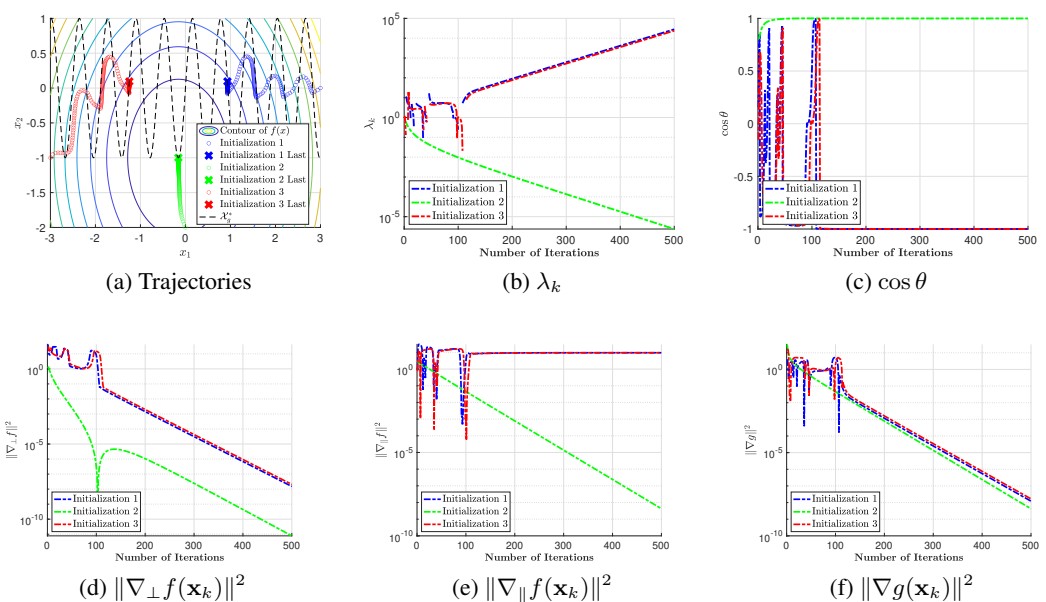

(a) Trajectories  (b) $\lambda_k$  (c) $\cos \theta$

(d) $\|\nabla_\perp f(\mathbf{x}_k)\|^2$  (e) $\|\nabla_\| f(\mathbf{x}_k)\|^2$  (f) $\|\nabla g(\mathbf{x}_k)\|^2$

Figure 3: Solving Problem (24) with different Initializations

In this additional experiment, we analyze the exact stationary points to which DBGD converges and examine the effect of different $\lambda_k$ values at these points, as discussed in Section 3.

We consider the problem in Equation (24) from Section 6 and run DBGD with $\phi(\mathbf{x}) = \|\nabla g(\mathbf{x})\|^2$ on the specified instance. As shown in Figure 3, the algorithm converges to three distinct stationary points, depending on the initialization. This behavior corresponds to the two scenarios discussed in Section 3, further supporting our theoretical insights.

- **Case I:** For Initialization 2 (green), DBGD converges to a point where both $\|\nabla f(\mathbf{x}_k)\|$ and $\|\nabla g(\mathbf{x}_k)\|$ are small. As shown in Figure 3(d), (e), and (f), all three metrics decrease. As illustrated in Figure 3(c), the cosine of the angle between $\nabla f(\mathbf{x}_k)$ and $\nabla g(\mathbf{x}_k)$ remains positive and eventually approaches 1. Figure 3(b) shows that $\lambda_k$ decreases to 0, aligning with the closed-form expression (16).

- **Case II:** For Initializations 1 and 3 (blue and red), DBGD converges to stationary points where $\|\nabla g(\mathbf{x}_k)\|$ is small, as shown in Figure 3(f). Additionally, $\nabla f(\mathbf{x}_k)$ has minimal energy in directions orthogonal to $\nabla g(\mathbf{x}_k)$, as seen in Figure 3(d). The remaining energy of $\nabla f(\mathbf{x}_k)$ is entirely in the opposite direction of $\nabla g(\mathbf{x}_k)$, since $\|\nabla_\| f(\mathbf{x}_k)\|$ does not converge (Figure 3(e)), and the angle between $\nabla f(\mathbf{x}_k)$ and $\nabla g(\mathbf{x}_k)$ is close to $180°$, as shown in Figure 3(c). In this case, $\lambda_k$ cannot be bounded by an absolute constant, as depicted in Figure 3(b), which is also consistent with our theoretical results.

## F.2 Additional Baselines

While there are no existing methods specifically tailored to the nonconvex simple bilevel setting, we include two additional baselines—BigSAM [19] and a-IRG [9]—which are originally designed for the convex case. We briefly review the update rules of these two algorithms below.

BigSAM is given by

$$\mathbf{y}_{k+1} = \mathbf{x}_k - \eta_g \nabla g(\mathbf{x}_k),$$
$$\mathbf{z}_{k+1} = \mathbf{x}_k - \eta_f \nabla f(\mathbf{x}_k),$$
$$\mathbf{x}_{k+1} = \alpha_{k+1}\mathbf{z}_{k+1} + (1 - \alpha_{k+1})\mathbf{y}_{k+1},$$

where $\eta_f$ and $\eta_g$ are constant stepsizes, and $\alpha_k = \min\{\frac{\gamma}{k}, 1\}$ for some $\gamma > 0$.

a-IRG is given by

$$\mathbf{x}_{k+1} = \mathbf{x}_k - \gamma_k(\nabla g(\mathbf{x}_k) + \eta_k \nabla f(\mathbf{x}_k)),$$

where $\gamma_k = \frac{\gamma_0}{\sqrt{k+1}}$ and $\eta_k = \frac{\eta_0}{(k+1)^{1/4}}$ for some constants $\gamma_0$ and $\eta_0$.

Following the same setup as in our original paper, we report the final gradient norms and $\cos\theta$ after 1000 iterations. The table below summarizes the performance of the considered algorithms in the first experiment, with parameters chosen via grid search.

| Method | Final $\|\nabla g\|^2$ | Final $\|\nabla_\perp f\|^2$ | Final $\cos(\theta)$ |
|---|---|---|---|
| DBGD | $8.5657 \times 10^{-17}$ | $1.0596 \times 10^{-16}$ | $-1.0000$ |
| Penalty $\lambda = 1$ | $1.8142$ | $4.4409 \times 10^{-16}$ | $1.0000$ |
| Penalty $\lambda = 10$ | $2.4473 \times 10^{-1}$ | $1.9800 \times 10^1$ | $0.2813$ |
| Penalty $\lambda = 100$ | $6.3210 \times 10^{-3}$ | $9.4788$ | $0.8692$ |
| Penalty $\lambda = 1000$ | $7.2475 \times 10^{-5}$ | $1.7477 \times 10^1$ | $0.7334$ |
| BigSAM | $2.7177 \times 10^{-4}$ | $2.1026 \times 10^1$ | $0.4741$ |
| a-IRG | $9.3776 \times 10^{-7}$ | $1.9802 \times 10^1$ | $0.6903$ |

Table 1: Toy Example

For the second experiment, we present the averaged results for each method across these parameter settings in the tables below. The total number of iterations is set to $10^6$.

| Method | $\|\nabla g\|$ | $\|\nabla_\perp f\|$ | $g(U)$ | $f(U)$ |
|---|---|---|---|---|
| DBGD | $5.59 \times 10^{-3}$ | $4.72 \times 10^{-1}$ | $2.73 \times 10^{-7}$ | $129.32$ |
| Penalty | $9.79 \times 10^{-1}$ | $1.06$ | $1.82 \times 10^{-2}$ | $134.67$ |
| BigSAM | $4.54 \times 10^{-3}$ | $5.71$ | $3.96 \times 10^{-4}$ | $134.80$ |
| a-IRG | $1.89 \times 10^{-2}$ | $1.81$ | $1.30 \times 10^{-4}$ | $135.40$ |

Table 2: Matrix Factorization $f_1$

| Method | $\|\nabla g\|$ | $\|\nabla_\perp f\|$ | $g(U)$ | $f(U)$ |
|---|---|---|---|---|
| DBGD | $7.12 \times 10^{-3}$ | $3.95 \times 10^{-1}$ | $8.15 \times 10^{-7}$ | $37.042$ |
| Penalty | $1.109$ | $2.23$ | $3.57 \times 10^{-2}$ | $48.543$ |
| BigSAM | $4.55 \times 10^{-3}$ | $7.72$ | $3.96 \times 10^{-4}$ | $51.254$ |
| a-IRG | $2.43 \times 10^{-2}$ | $2.75$ | $1.05 \times 10^{-4}$ | $52.709$ |

Table 3: Matrix Factorization $f_2$

As shown in the tables, it is not surprising that BiG-SAM and a-IRG underperform compared to DBGD in terms of our proposed stationarity metrics, as they are not specifically designed for the nonconvex setting. In particular, their performance is similar to that of the penalty method with a large penalty parameter—overemphasizing the lower-level objective while failing to adequately control the upper-level. The failure of algorithms designed for convex simple bilevel optimization when applied to nonconvex simple bilevel problems highlights the necessity of studying the nonconvex setting.

### F.3 Other Applications for Simple Bilevel Optimization

Simple bilevel optimization arises in various applications, such as sparse representation learning [3], fairness regularization [2], and dictionary learning [4]. In what follows, we illustrate several specific formulations.

**Sparsity Representation Learning.** We learn sparse feature representations on a supervised dataset $\mathcal{D}$ of $(\mathbf{x}, y)$ pairs by applying a non-convex $L_p$ regularization:

$$f(\theta) = \mathbb{E}_{\mathcal{D}}\big[\ell\big(y,\ \phi_\theta\big(h_\theta(x)\big)\big)\big], \qquad g(\theta) = \mathbb{E}_{\mathcal{D}}\big[\|h_\theta(x)\|_p^p\big],$$

where $\ell(\cdot, \cdot)$ is the data loss, $h_\theta(x) \mapsto z \in \mathbb{R}^m$ is a hidden feature map, $\phi_\theta$ is a prediction head, and $p$ is a power coefficient.

**Fairness Classification.** Concretely, the lower-level problem is a sparse logistic-regression problem for some $\lambda > 0$:

$$\min_{\beta \in \mathbb{R}^d}\ g(\beta) = -\frac{1}{n} \sum_{i=1}^{n} \log \mathbb{P}\big(\hat{y}_i = y_i \mid x_i; \beta\big) \quad \text{s.t.} \quad \|\beta\|_1 \leq \lambda,$$

while the upper-level objective is the squared covariance:

$$f(\beta) = \left(\frac{1}{n} \sum_{i=1}^{n} (v_i - \bar{v})\, \mathbb{P}\big(\hat{y}_i = 1 \mid x_i; \beta\big)\right)^2.$$

**Dictionary Learning.** We aim to find the dictionary $\tilde{D} \in \mathbb{R}^{m \times q}$ ($q > p$) and the coefficient matrix $\tilde{X} \in \mathbb{R}^{q \times n'}$ for the new dataset $A'$, and at the same time enforce $\tilde{D}$ to perform well on the old dataset $A$ together with the learned coefficient matrix $\tilde{X}$. This leads to the following bilevel problem:

$$\min_{\tilde{D} \in \mathbb{R}^{m \times q}}\ \min_{\tilde{X} \in \mathbb{R}^{q \times n'}}\ f(\tilde{D}, \tilde{X})$$
$$\text{s.t.} \qquad \|\tilde{x}_k\|_0 \leq \delta, \quad k = 1, \ldots, n',$$
$$\tilde{D} \in \arg\min_{\|\tilde{d}_j\|_2 \leq 1} g(\tilde{D}),$$

where the objective

$$f(\tilde{D}, \tilde{X}) \triangleq \frac{1}{2n'} \sum_{k=1}^{n'} \big\|a'_k - \tilde{D}\, \tilde{x}_k\big\|_2^2$$

is the average reconstruction error on the new dataset $A'$, and the lower-level objective

$$g(\tilde{D}) \triangleq \frac{1}{2n} \sum_{i=1}^{n} \big\|a_i - \tilde{D}\, \tilde{x}_i\big\|_2^2$$

is the error on the old dataset $A$.

