# OpenReview forum: "On the Complexity of Finding Stationary Points in Nonconvex Simple Bilevel Optimization"
_NeurIPS.cc/2025/Conference — NeurIPS 2025 poster_

### Official Review · Reviewer_qGQn · 2025-07-01

**Clarity:** 2
**Significance:** 3
**Originality:** 3
**Rating:** 4
**Confidence:** 4

**Summary:**

This paper studied a smooth nonconvex simple bilevel optimization problem and developed the stationary condition for it. They then proved that dynamic barrier gradient descent algorithm can converge to the defined stationary point in polynomial time. Specifically, to reach an $(\epsilon_f, \epsilon_g)$ stationary point, the algorithm needs $\max(\epsilon_f^{-\frac{3+p}{1+p}},\epsilon_f^{-\frac{3+p}{2}})$ iterations. Numerical experiments validate the effectiveness of the proposed algorithm.

**Questions:**

1. Do you intend to show that the proposed stationarity condition implies (6) and (7)? If so, please provide a more detailed explanation.
2. The domain of $\lambda$ is unclear. I recommend distinguishing between cases where $\lambda$ takes a finite value, i.e., $\lambda \in \mathbb{R}_+$, and cases where it may include infinity, i.e., $\lambda \in (0, +\infty]$. For example, in the line below Line 168, the KKT multiplier $\lambda$ should be finite. However, in equation (3) and the algorithm design, $\lambda$ can be infinite.
3. Please refer to Weaknesses 1) and 2) for concerns related to Lemma 3.1.

**Ethical Concerns:**

["NO or VERY MINOR ethics concerns only"]

**Final Justification:**

This paper established the stationary condition for nonconvex smooth simple bilevel optimization without any additional assumptions on lower-level problem such as Polyak-Lojaswicz condition. This contribution is significant to bilevel optimization because finding an appropriate stationary condition serves as a foundation to establish algorithm convergence. This paper also analyzed the convergence rate of dynamic barrier gradient descent algorithm under this setting. Since all of my concerns are solved, I increased the score to 4.

**Limitations:**

The main limitation lies in the potential overstatement of the conclusions drawn from Lemma 3.1.

**Quality:**

3

**Strengths And Weaknesses:**

Strengths: 1) The problem studied in this paper is significant for both simple bilevel optimization and general bilevel optimization. By removing convexity and PL conditions at the lower level, the authors provide a broader theoretical guarantee for stationarity. 2) The idea of decomposing the upper-level gradient into orthogonal and parallel components is interesting and innovative. 3) The proposed algorithm is supported by a strong convergence guarantee.

Weakness: (1) The main concern is the correctness—or perhaps overstatement—of Lemma 1. For a smooth function, if the local region is chosen to be sufficiently small (as in the proof, where the radius $r$ is on the order of the target error, i.e., very small), then almost any point could qualify as the "near-local optimum" under the authors’ definition. This is especially problematic when the definition’s validity depends heavily on $r$. Additionally, the conclusion of Lemma 1 seems counterintuitive: for a smooth function, a small gradient does not rule out the possibility of being a local maximum. From my perspective, a more robust definition of a near-local minimum should require the existence of a fixed-radius local region where the inequalities in Lemma 3.1 hold.

(2) Another weakness is that I do not understand the second point of Lemma 3.1. It is rigorous to say that for any point with improvement on the lower-level objective, the upper-level objective changes is not to much since $\delta$ is very small. But I do not see why the improvement on the lower-level will lead to the negative effect on the upper-level, as said in this paper. I quoted the statement here "the upper-level objective cannot be significantly improved without negatively impacting".

(3) The above two concerns is on the definition of stationarity. I also do not understand the relations of existing stationary measure in (6), especially there is a strange (ii) in Line 176.

---

> ### Author Rebuttal · Authors · 2025-07-31
>
> **W1. The main concern is the correctness—or perhaps overstatement—of Lemma 1. For a smooth function, if the local region is chosen to be sufficiently small (as in the proof, where the radius $r$ is on the order of the target error, i.e., very small), then almost any point could qualify as the "near-local optimum" under the authors’ definition. This is especially problematic when the definition’s validity depends heavily on $r$. Additionally, the conclusion of Lemma 1 seems counterintuitive: for a smooth function, a small gradient does not rule out the possibility of being a local maximum. From my perspective, a more robust definition of a near-local minimum should require the existence of a fixed-radius local region where the inequalities in Lemma 3.1 hold.**
>
> **R1.**
> Thank you for raising this question. We believe there may be a misunderstanding regarding the first condition in Lemma 3.1. Even if the local region is chosen to be sufficiently small, it is *not* the case that almost any point would satisfy this condition. Specifically, suppose $g$ is $G$-Lipschitz; then for any $x$, we have $g(x) \geq g(\hat{x})-G\\|\hat{x}-x\\|$. However, our condition in Lemma 3.1 is much stronger: it requires $g(x) \geq g(\hat{x})-(1+\delta)\sqrt{\epsilon_g}\\|\hat{x}-x\\|$, which replaces the Lipschitz constant $G$ with the much smaller coefficient $\sqrt{\epsilon_g}$. As a concrete example, consider $g(x) = x^2$. When
> $x$ is sufficiently large, this condition cannot be satisfied no matter how small the local region is.
>
> The reviewer is correct that our definition does not preclude $\hat{x}$ from being a local maximum. Indeed, it measures approximate stationarity via the gradient norm (Definition 3.1), which is standard in nonconvex optimization. Note that our conclusion following Lemma 3.1 remains valid even if $\hat{x}$ is a local maximum, since due to smoothness, the function value cannot decrease significantly within a small neighborhood in that case either. We also agree that it would be a stronger notion of a near-local minimum if the radius could be chosen independently of the target error $\epsilon$. However, achieving this appears to be a nontrivial task, even in standard single-level minimization problems.
>
> Lastly, to make Lemma 3.1 more precise, such a radius should be treated as an upper bound. Specifically, a point $\hat{x} \in \mathbb{R}^n$ is an $(\epsilon_f, \epsilon_g)$-stationary point of SBO if and only if the following holds: for any $\delta > 0$, there exists a radius $\hat{r} > 0$ such that for all $0 < r \leq \hat{r}$, both conditions in Lemma 3.1 are satisfied. This refinement does not affect the validity of the proof, and such a radius $\hat{r}$ can be determined based on the analysis provided in Appendix A.
>
> ---
>
> **W2. Another weakness is that I do not understand the second point of Lemma 3.1. It is rigorous to say that for any point with improvement on the lower-level objective, the upper-level objective changes is not to much since $\delta$ is very small. But I do not see why the improvement on the lower-level will lead to the negative effect on the upper-level, as said in this paper. I quoted the statement here "the upper-level objective cannot be significantly improved without negatively impacting".**
>
> **R2.**
> Thank you for the comment.
> Specifically, when the second condition is satisfied, then for any $ \\|x - \hat{x}\\| \leq r $, if
> $
> f(x) < f(\hat{x}) - (1 + \delta)\sqrt{\epsilon_f} \\|x - \hat{x}\\|
> $ (i.e., if the upper-level objective is significantly improved),
> it must be that $ g(x) > g(\hat{x}) $ (meaning the lower level objective is negatively impacted).
> In this sense, it indicates that we aim to find points where the upper-level objective cannot be significantly improved without negatively affecting the lower-level objective. If such a point is found, approximate stationarity is achieved.
>
> We are not asserting that this is always the case---if both objectives can be decreased simultaneously, the point is not stationary by our definition. Therefore, the two conditions in Lemma 3.1 should be interpreted jointly, rather than viewed separately.
>
> ---
>
> **W3. The above two concerns is on the definition of stationarity. I also do not understand the relations of existing stationary measure in (6), especially there is a strange (ii) in Line 176.**
>
> **R3.** Thank you for the question. We would like to clarify that metrics (5) and (6) are not our proposed metrics; they are convergence metrics commonly used in the constrained optimization literature [1, 2]. As noted in lines 172–176, these works aim to find a point $x$ that satisfies either (5) or (6). In particular, (6) indicates that such a point is stationary but not necessarily globally optimal for $g$. We further elaborate on its connection with our proposed metric in response to your next question below.
>
> ---
>
> **Q1. Do you intend to show that the proposed stationarity condition implies (6) and (7)? If so, please provide a more detailed explanation.**
>
> **A1.** Yes. A point satisfying either (6) or (7) is considered stationary in the constrained optimization literature [3]. As noted in lines 178–181, we state that if a point $x$ satisfies our proposed metric, then $x$ also satisfies either (6) or (7), which can be verified directly.
>
> Specifically, suppose $x$ is an $(\epsilon_d^2,\epsilon_d^2)$-stationary point (Definition 3.1). If $ g(x) - g^* \geq 0.99 \epsilon_{p} $, condition (6) follows from the first condition in Definition 3.1; otherwise, $g(x) - g^* < 0.99 \epsilon_{p} < \epsilon_p$, and condition (7) holds due to the second condition in Definition 3.1.
>
> Moreover, since condition (7) (the unscaled KKT condition) is stronger than condition (5) (the scaled KKT condition), it follows that if $ x $ is an $(\epsilon_d^2,\epsilon_d^2)$-stationary point, then $x$ satisfies either (5) or (6) as well.
>
> ---
>
> **Q2. The domain of $\lambda$ is unclear. I recommend distinguishing between cases where $\lambda$ takes a finite value, i.e., $\lambda \in \mathbb{R}_+$, and cases where it may include infinity, i.e., $\lambda\in(0, +\infty]$. For example, in the line below Line 168, the KKT multiplier should be finite. However, in equation (3) and the algorithm design,
>  can be infinite.**
>
>  **A2.** Thank you for the question. To clarify, in both our proposed definition and algorithm design, $\lambda$ is always assumed to be finite,  although we do not impose a uniform upper bound on its value. We will add this point to the paper.
>
>  ---
>
> References:
>
> [1]. Coralia Cartis, Nicholas IM Gould, and Philippe L Toint. On the complexity of finding first order critical points in constrained nonlinear optimization. Mathematical Programming, 2014.
>
> [2]. Coralia Cartis, Nicholas IM Gould, and Ph L Toint. Corrigendum: On the complexity of finding first-order critical points in constrained nonlinear optimization. Mathematical Programming, 2017
>
> [3]. Ernesto G Birgin, JL Gardenghi, José Mario Martínez, Sandra A Santos, and Ph L Toint. Evaluation complexity for nonlinear constrained optimization using unscaled kkt conditions and high-order models. SIAM Journal on Optimization, 2016.

---

> ### Comment · Reviewer_qGQn · 2025-08-03
>
> I thank the authors for the detailed response. It solves most of my questions, especially W1 and W2. I have the remaining question on $\lambda$ and corresponding CQ condition required for condition (5)--(6). I understand that for the $\lambda_k$ using in the algorithm, each $\lambda_k$ is finite and you do not put an upper bound for $\lambda$ sequence. But if you consider the limit of the algorithm when $K\rightarrow \infty$, the optimal $\lambda_{k^*}$ seems should be infinite, which does not align with the current Definition 3.1. Moreover, it is known that Definition 3.1 (or (6)) with finite $\lambda$ usually does not hold for bilevel problems, or only holds under specific constraint qualifications (CQs); see, for example, [1,2]. Therefore, how can such settings be excluded by Definition 3.1?
>
> One possible solution seems to be ensuring that the $\lambda$ defined in Definition 3.1 is negatively correlated with the target error $\epsilon_f$.
>
>
> [1]. Gong et al. Bi-objective trade-off with dynamic barrier gradient descent.
>
> [2]. Xiao et al. A Generalized Alternating Method for Bilevel Learning under the Polyak-{\L} ojasiewicz Condition.

---

> > ### Author Response · Authors · 2025-08-04
> >
> > Thank you very much for engaging in the discussion and the follow-up question. We agree with the reviewer that the sequence $\{\lambda_k\}$ may go to infinity asymptotically, which can be a potential issue for Definition 3.1.
> > Specifically, as $K\rightarrow \infty$, the algorithm may converge to a point $x_{\infty}$ where $\nabla g(x_{\infty})=0$, in which case there may not be a finite $\lambda$ such that $x_{\infty}$ and $\lambda$ satisfy the second condition in Definition 3.1.
> >
> > However, in this limiting case,  the above issue can be addressed by considering the alternative stationarity condition in Definition 3.2 (KKT condition of the gradient-based reformulation). In particular, the second condition in Definition 3.1 is replaced by $\\|\nabla f(\hat{x}) + \nabla^2 g(\hat{x})w\\|^2 \leq \epsilon_p$ for some bounded vector $w$. Note that our algorithm ensures that $\lambda_k =\mathcal{O}(1 / \\|\nabla g({x}_k)\\|)$, so the product $\lambda_k \\|\nabla g({x}_k)\\|$ remains bounded and has a finite limit point. By applying Theorem 3.2, we can show that the limit point of our algorithm satisfies Definition 3.2 with $\epsilon_p = O(\epsilon_f + \epsilon_g)$ under Assumption 3.1.
> >
> > Finally,  we note that reaching a point with exactly vanishing gradient is rare in practice, and since $\lambda_k = \mathcal{O}(1/||\nabla g(x_k)||)$, the sequence ${\lambda_k}$ remains finite in all practical cases. We would add a remark to clarify this point in the revision.

---

> ### Comment · Reviewer_qGQn · 2025-08-04
>
> I thank the author for this clarification. Since all of my concerns have been solved, I will increase the score to 4.

---

### Official Review · Reviewer_ygXd · 2025-07-01

**Clarity:** 4
**Significance:** 3
**Originality:** 3
**Rating:** 4
**Confidence:** 3

**Summary:**

This paper studies a simple bilevel optimization problem where both the upper- and the lower-level problems are non-convex. First, the authors introduce a novel notion of stationarity, tailored to the specific problem at hand, and establish its relationship to existing metrics in the bilevel optimization literature. Then, they apply the Dynamic Barrier Gradient Descent (DBGD) algorithm (originally developed for constrained optimization problems) to the nonconvex simple bilevel problem and establish discrete-time stationarity guarantees. We note that this represents the first discrete-time complexity bound that guarantees stationarity at both levels in such problems. Finally, a set of numerical experiments on toy examples and matrix factorization tasks illustrate DBGD method's superior performance over traditional penalty methods.

**Questions:**

- To situate the method within the existing literature, a table summarizing relevant results about simple bilevel optimization problems (method name, assumptions, optimality metric, complexity, etc.) could be helpful.
- Is Assumption 3.1 (Local Error Bound) typically used in similar settings (ie simple nonconvex bilevel problems)? Is it a strong assumption? Are there common applications in which this assumptions holds?
- The definition of stationarity for the proposed problem is a central contribution of this work and should be properly situated within the relevant literature. Has this particular stationarity metric, or a related one, been previously introduced or discussed in the bilevel optimization literature?
- What approach did you use to tune the penalty parameter (of the penalty method baseline in the experiments) for obtaining the best possible performance and ensuring fair comparisons?

**Ethical Concerns:**

["NO or VERY MINOR ethics concerns only"]

**Final Justification:**

The authors addressed my questions. I’m generally satisfied with the response and recommend acceptance of the paper. Specifically, the authors plan to include a discussion in the revised paper on the applications of their problem, and they will expand the experimental section by adding more baselines. However, I still believe a rating of 4 is appropriate.

**Limitations:**

Yes

**Quality:**

4

**Strengths And Weaknesses:**

Strengths
- This paper deals with a challenging bilevel problem. It is a nonconvex simple bilevel problem where both the upper and lower-level objectives are nonconvex functions.
- A suitable stationarity concept is introduced for the specific nonconvex simple bilevel problem, and its relationship to existing metrics in the bilevel optimization literature is established.
- The paper provides the first discrete-time complexity bound that guarantees stationarity at both levels of a  nonconvex simple bilevel problem.

Weaknesses
- Since this is a simple bilevel problem, its applicability is somewhat limited. In fact, the only application discussed in detail is matrix factorization. The authors may wish to provide more information about the potential applications of the specific bilevel problem they are addressing, e.g. by providing specific example applications and their formulations.
- The experimental evaluation is somehow limited. There is only one baseline, the penalty method and the experiments consist of toy problems and small scale matrix factorization tasks. Are there other bilevel methods that can be used for the solution of this problem? How about methods developed for generic (not necessarily) simple bilevel problems?

---

> ### Author Rebuttal · Authors · 2025-07-31
>
> **W1. Since this is a simple bilevel problem, its applicability is somewhat limited. In fact, the only application discussed in detail is matrix factorization. The authors may wish to provide more information about the potential applications of the specific bilevel problem they are addressing, e.g. by providing specific example applications and their formulations.**
>
>
> **R1.** Thank you for your comment!
> Although the paper focuses on results on matrix factorization, simple bilevel optimization has many other applications. These include sparse representation learning, fairness regularization, and lexicographic optimization [1]; over-parameterized regression and dictionary learning [2, 3]; and linear inverse problems [4].
>
> Below, we provide some specific formulations:
>
> *Sparse Representation Learning.* We learn sparse feature representations on a supervised dataset $\mathcal{D}$ of $(x,y)$ pairs by applying a non-convex $L_p$ regularization:
>
> $$
> f(\theta) = \mathbb{E} [\ell(y, \phi_{\theta}(h_{\theta}(x)))], \quad
> g(\theta) = \mathbb{E} [\|h_{\theta}(x)\|_{p}^{p}]
> $$
>
>
> where $\ell(\cdot,\cdot)$ is the data loss, $h_{\theta}(x) \mapsto z \in \mathbb{R}^{m}$ is a hidden feature map, $\phi_{\theta}$ is a prediction head, and $p$ is a power coefficient.
>
> *Fairness Classification.*
> Concretely, the lower-level problem is a logistic regression problem:
>
> $$
> g(\beta) = -\frac{1}{n} \sum_{i=1}^{n} \log \mathbb{P}(\hat{y}_i = y_i \mid \mathbf{x}_i; \beta)
> $$
>
> while the upper-level objective is the squared covariance:
>
> $$
> f(\beta) = \left( \frac{1}{n} \sum_{i=1}^{n} (v_i - \bar{v}) \mathbb{P}(\hat{y}_i = 1 \mid \mathbf{x}_i; \beta) \right)^{2}
> $$
>
> *Dictionary Learning.* We aim to find the dictionary $\tilde{\mathbf{D}} \in \mathbb{R}^{m \times q}$ ($q > p$) and the coefficient matrix $\tilde{\mathbf{X}} \in \mathbb{R}^{q \times n'}$ for the new dataset $A'$, and at the same time enforce $\tilde{\mathbf{D}}$ to perform well on the old dataset $A$ together with the learned coefficient matrix $\hat{\mathbf{X}}$. This leads to the following bilevel problem:
> $$
> f(\tilde{\mathbf{D}}, \tilde{\mathbf{X}}) \triangleq \frac{1}{2n'} \sum_{k=1}^{n'} \| \mathbf{a}'_{k} - \tilde{\mathbf{D}} \tilde{\mathbf{x}}_k \|_2^2
> $$
>
> is the average reconstruction error on the new dataset $A'$, and the lower-level objective
>
> $$
> g(\tilde{\mathbf{D}}) \triangleq \frac{1}{2n} \sum_{i=1}^{n} \| \mathbf{a}_{i} - \tilde{\mathbf{D}} \hat{\mathbf{x}}_i \|_2^2
> $$
>
> is the error on the old dataset $A$.
> We will include this discussion in the revised paper.
>
> ---
>
> **W2. The experimental evaluation is somehow limited. There is only one baseline, the penalty method and the experiments consist of toy problems and small scale matrix factorization tasks. Are there other bilevel methods that can be used for the solution of this problem? How about methods developed for generic (not necessarily) simple bilevel problems?**
>
> **R2.** Thanks for the question. To the best of our knowledge, there are no other existing methods for solving non-convex simple bilevel optimization with theoretical guarantees. Our result is the first complexity result for a discrete-time algorithm that guarantees joint stationarity for both levels in nonconvex simple bilevel problems.
>
> Regarding methods for solving general bilevel optimization problems, as discussed in Appendix D (lines 905–920), we highlight the connections between algorithms designed for simple bilevel optimization and those for general bilevel optimization. Specifically, when applied to the simple bilevel setting (where no upper-level variable is present), most existing methods for general bilevel optimization reduce to the penalty method. This is the reason we selected the penalty method as our baseline.
>
> Furthermore, we also include additional baselines—algorithms originally developed for the convex case—which further demonstrate that DBGD outperforms them. Please refer to R1 to Reviewer oo1n for more details.
>
> ---
>
> **Q1. To situate the method within the existing literature, a table summarizing relevant results about simple bilevel optimization problems (method name, assumptions, optimality metric, complexity, etc.) could be helpful.**
>
> **A1.** Thank you for the question. To the best of our knowledge, there are no other methods that provably solve nonconvex simple bilevel optimization. Our work provides the first complexity result for a discrete-time algorithm that guarantees joint stationarity for both levels in general nonconvex simple bilevel problems. Regarding the convergence metric, we offer a detailed discussion comparing the guarantees in general bilevel optimization and constrained optimization in Section 3.1, and we also elaborate on this below in response to Q3.
>
> Additionally, we discuss related work on convex simple bilevel optimization in Section 1.1. If necessary, we can provide a consolidated summary of methods in a table for solving both convex and nonconvex simple bilevel problems.
>
> ---
>
> **Q2. Is Assumption 3.1 (Local Error Bound) typically used in similar settings (ie simple nonconvex bilevel problems)? Is it a strong assumption? Are there common applications in which this assumptions holds?**
>
> **A2.**
> Since nonconvex simple bilevel problems are underexplored, there are very few works that have even considered this setting. The local error bound condition is widely used in the constrained optimization literature [3, 4]. We believe this assumption is not strong, as it can be implied by a local PL inequality, which is itself a relaxation of the global PL condition—verified, for instance, for overparameterized least-squares and certain neural network loss functions [5].
>
> ---
>
> **Q3. The definition of stationarity for the proposed problem is a central contribution of this work and should be properly situated within the relevant literature. Has this particular stationarity metric, or a related one, been previously introduced or discussed in the bilevel optimization literature?**
>
> **A3.** To the best of our knowledge, no stationarity metric has previously been proposed for this setting. However, related stationarity metrics have been studied in constrained optimization and general bilevel optimization. In Section 3.1, we provide a detailed comparison of the guarantees in these settings. Specifically, our proposed metric implies the commonly used scaled and unscaled KKT conditions in constrained optimization (as shown in Section 3.1.1) and also aligns with a widely adopted metric, KKT conditions for gradient-based reformulation, in general bilevel optimization, as explicitly stated in Theorem 3.2.
>
> ---
>
> **Q4. What approach did you use to tune the penalty parameter (of the penalty method baseline in the experiments) for obtaining the best possible performance and ensuring fair comparisons?**
>
> **A4.** Thanks for the question. We use a grid search, which is a common practice for hyperparameter tuning. Specifically, we explored the penalty parameter
> $\lambda \in \\{ 10^{-k} \mid k \in [-5, 5] \\}$, and found that the performance is better for $k = \\{0, 1, 2, 3\\}$.
>
>
> ---
>
>
>
> References:
>
> [1]. Chengyue Gong, Xingchao Liu, and Qiang Liu. Bi-objective trade-off with dynamic barrier
> 371 gradient descent. In Proceedings of the 35th International Conference on Neural Information
> 372 Processing Systems, pages 29630–29642, 2021.
>
> [2]. Ruichen Jiang, Nazanin Abolfazli, Aryan Mokhtari, and Erfan Yazdandoost Hamedani. A conditional gradient-based method for simple bilevel optimization with convex lower-level problem. In International Conference on Artificial Intelligence and Statistics, pages 10305–10323. PMLR, 2023.
>
> [3]. Jincheng Cao, Ruichen Jiang, Nazanin Abolfazli, Erfan Yazdandoost Hamedani, and Aryan Mokhtari. Projection-free methods for stochastic simple bilevel optimization with convex lower-level problem. Advances in Neural Information Processing Systems, 36, 2023.
>
> [4]. Shoham Sabach and Shimrit Shtern. A first order method for solving convex bilevel optimization problems. SIAM Journal on Optimization,27(2):640–660, 2017.
>
> [5]. Xu, Yi, Qihang Lin, and Tianbao Yang. "Accelerated stochastic subgradient methods under local error bound condition." arXiv preprint arXiv:1607.01027 (2016).
>
> [6]. Liu, Hongwei, Ting Wang, and Zexian Liu. "Convergence rate of inertial forward–backward algorithms based on the local error bound condition." IMA Journal of Numerical Analysis 44.2 (2024): 1003-1028.
>
> [7]. Liu, C., Zhu, L., and Belkin, M. Loss landscapes and optimization in over-parameterized non-linear systems and neural networks. Applied and Computational Harmonic Analysis, 59:85–116, 2022.

---

> > ### Comment · Reviewer_ygXd · 2025-08-01
> >
> > Thank you for your response. I have no further questions at this point.

---

### Official Review · Reviewer_JLoC · 2025-07-02

**Clarity:** 4
**Significance:** 4
**Originality:** 4
**Rating:** 5
**Confidence:** 1

**Summary:**

This paper studies the complexity of finding stationary points in nonconvex simple bilevel optimization problems, where the upper-level objective is minimized over the solution set of a lower-level problem. The authors introduce a notion of \((\epsilon_f, \epsilon_g)\)-stationarity to measure convergence, capturing points where the upper-level objective cannot be improved without worsening the lower-level objective. They adopt a variant of the Dynamic Barrier Gradient Descent (DBGD) algorithm and establish the first discrete-time complexity guarantee for nonconvex simple bilevel problems, showing that DBGD achieves an \((\epsilon_f, \epsilon_g)\)-stationary point in \(\mathcal{O}(\max(\epsilon_f^{-\frac{3+p}{1+p}}, \epsilon_g^{-\frac{3+p}{2}}))\) iterations, where p \geq 0 is a tunable parameter. Furthermore numerical experiments validate the algorithm's effectiveness compared to penalty-based methods.

**Questions:**

NA

**Ethical Concerns:**

["NO or VERY MINOR ethics concerns only"]

**Final Justification:**

Authors have answered my questions.

**Quality:**

4

**Strengths And Weaknesses:**

Strengths:

This is a well-written paper that makes a valuable contribution by extending the framework of [46] to a non-convex setting for the low-level function. The authors provide a complete algorithmic solution and rigorously establish convergence guarantees, demonstrating both theoretical and practical advancements over prior work.

Weaknesses:

Could the authors elaborate on the additional technical difficulties arising from the non-convexity of the low-level function in their framework, compared to the convex formulation in [46]? Specifically:
1) How does non-convexity affect the theoretical analysis (e.g., convergence guarantees, optimality conditions)?

2) What algorithmic modifications were required to address these challenges?

3) Are there measurable performance trade-offs between the two approaches in practice?

---

> ### Author Rebuttal · Authors · 2025-07-31
>
> **W. Could the authors elaborate on the additional technical difficulties arising from the non-convexity of the low-level function in their framework, compared to the convex formulation in [46]?**
>
> Thank you for the question. The algorithm in [46] is designed for general bilevel optimization and requires the lower-level objective to be strongly convex. In contrast, we consider a slightly different setting—known as simple bilevel optimization—where there is no implicit dependency between the upper and lower-level variables, and the lower-level problem is generally nonconvex and smooth without additional assumptions.
>
> Within the context of simple bilevel optimization, several works have focused on the convex setting, such as [1], [2], and [3]. Among these, [1] presents the most similar algorithm, and we demonstrate that a variant of it can be viewed as belonging to the same family as DBGD, as shown in Appendix C. We now explain the challenges arising from the nonconvexity of the lower level and address your specific questions below.
>
> **W1. How does non-convexity affect the theoretical analysis (e.g., convergence guarantees, optimality conditions)?**
>
> **R1.** Unlike simple bilevel optimization with a convex lower level, nonconvex simple bilevel optimization remains largely underexplored. In particular, no prior work has proposed optimality or stationarity conditions specifically for the nonconvex simple bilevel setting. Due to the lack of convexity in the lower level, the lower-level solution set can be nonconvex, making it intractable to achieve any form of global optimality. Consequently, as in its single-level counterpart, the primary objective in nonconvex simple bilevel optimization is to find a near-stationary point rather than a near-optimal solution, as defined in [1], [2], and [3]. In Section 3, we introduce a stationarity metric by decomposing the upper-level gradient into components parallel and orthogonal to the lower-level gradient, and we provide a rigorous interpretation of this metric in Lemma 3.1.
>
> Regarding convergence guarantees, we show that a variant of DBGD [4] can solve nonconvex simple bilevel optimization up to joint stationarity for both the upper and lower levels. To the best of our knowledge, this is the first complexity result for a discrete-time algorithm that establishes such joint stationarity guarantees for general nonconvex simple bilevel problems.
>
> ---
>
> **W2. What algorithmic modifications were required to address these challenges?**
>
> **R2.** Similar algorithms for solving convex simple bilevel optimization problems [1, 3] use cutting-plane methods to approximate the lower-level solution set. However, these approaches do not extend to the nonconvex setting, as the lower-level solution set may be nonconvex, making it difficult to approximate and impossible to ensure the true solution lies within the approximated set.
>
> As detailed in Appendix C, both a variant of the algorithm in [1] and our proposed algorithm can be viewed as instances of a shared framework, distinguished by their choice of the parameter function $\phi(x)$. To adapt this framework to the nonconvex setting, we choose $\phi(x) = \beta ||\nabla g(x)||^{2}$, in contrast to the convex setting where $\phi(x) = (g(x) - g^{*})/\eta$ is used. Here, $\phi(x)$ reflects the suboptimality of the point $x$. Given that global optimality is generally unattainable in the nonconvex case, it is natural to replace the optimality gap with the stationarity measure $||\nabla g(x)||^2$, which is commonly adopted in nonconvex optimization. This modification leads to the development of our analyzed algorithm and its corresponding analysis.
>
> ---
>
> **W3. Are there measurable performance trade-offs between the two approaches in practice?**
>
> **R3.**
> To assess measurable performance, we include additional baselines originally designed for convex simple bilevel problems. Please refer to R1 to Reviewer oo1n for details. As shown in the tables, these methods underperform compared to DBGD. This performance gap underscores the limitations of convex-based algorithms in the nonconvex setting and highlights the need for dedicated methods tailored to nonconvex simple bilevel optimization.
>
>
> ---
>
>
> References:
>
> [1]. Ruichen Jiang, Nazanin Abolfazli, Aryan Mokhtari, and Erfan Yazdandoost Hamedani. A conditional gradient-based method for simple bilevel optimization with convex lower-level problem. In International Conference on Artificial Intelligence and Statistics, pages 10305–368
> 10323. PMLR, 2023
>
> [2]. Sepideh Samadi, Daniel Burbano, and Farzad Yousefian. Achieving optimal complexity
> guarantees for a class of bilevel convex optimization problems. arXiv preprint arXiv:2310.12247,
> 2023.
>
> [3]. Jincheng Cao, Ruichen Jiang, Erfan Yazdandoost Hamedani, and Aryan Mokhtari. An accelerated gradient method for convex smooth simple bilevel optimization. Advances in Neural Information Processing Systems, 37:45126–45154, 2024.
>
> [4]. Chengyue Gong, Xingchao Liu, and Qiang Liu. Bi-objective trade-off with dynamic barrier
> 371 gradient descent. In Proceedings of the 35th International Conference on Neural Information
> 372 Processing Systems, pages 29630–29642, 2021.

---

### Official Review · Reviewer_oo1n · 2025-07-03

**Clarity:** 3
**Significance:** 3
**Originality:** 3
**Rating:** 4
**Confidence:** 4

**Summary:**

This paper studies the challenging problem of simple bilevel optimization where both the upper- and lower-level objective functions are smooth but potentially nonconvex. Because finding a global optimum in this setting is generally intractable, the work focuses on the more achievable goal of finding a near-stationary point. The authors first introduce a suitable notion of an $(\epsilon_f, \epsilon_g)$-stationary point, which intuitively identifies a solution where the upper-level objective cannot be substantially improved locally without causing a larger deterioration in the lower-level objective. This new metric consists of two conditions: the norm square of the lower-level gradient must be smaller than $\epsilon_g$, and the norm square of the upper-level gradient + $\lambda$ * the lower-level gradient must be smaller than $\epsilon_f$.

To find such a point, the paper analyzes a discrete-time variant of the Dynamic Barrier Gradient Descent (DBGD) algorithm (previously proposed by Gong et al.) The core of the method is a first-order update step where the descent direction is a dynamic combination of the two gradients, controlled by a multiplier $\lambda_k$ that is gradually increasing over iterations. The paper's main contribution is providing the first complexity result for a discrete-time algorithm that guarantees joint stationarity for both levels in this general nonconvex setting. Specifically, the algorithm is shown to find an $(\epsilon_f, \epsilon_g)$-stationary point with a complexity in $O(\max (\epsilon_f^{-\frac{3+p}{1+p}}, \epsilon_g^{-\frac{3+p}{2}}) )$.

I have checked most of the proofs in the main paper and the appendix. The main analysis is based on a cute application of the existing results from DBGD by Gong et al. Specifically, the results in Section 4 are the same as the derivation and results in Gong et al. in their Section 2, the analysis of DBGD. The major contribution of this paper starts in Section 5 by applying descent lemmas to both function $f$ and $g$ (Lemma 5.1). The discussion below Lemma 5.1 follows directly by weighting the terms in Lemma 5.1 and this leads to a $\lambda_k$-dependent bound on the gradient norm. However, since the Lagrangian dual $\lambda_k$ is increasing over iterations, this leads to additional discussion on Lemma 5.2 - 5.5 to ensure a $\lambda_k$-independent bound. The bound is achieved by playing with the inequality to remove the $\lambda_k$ term in Equation 17. This can be achieved by replacing $\lambda_k$ by its upper bound. The rest of the analysis follows by doing some linear algebra tricks then. I like the analysis in Section 5 and the experimental results in Section 6. The experimental results verify that the proposed algorithm can achieve low gradient norm for both the lower level and the upper level. The baselines for comparison are relatively weak as it only uses fixed penalty parameter. I wonder if there are better baselines that the authors can compare with.

**Questions:**

Please respond to my questions in the strengths and weaknesses section.

**Ethical Concerns:**

["NO or VERY MINOR ethics concerns only"]

**Final Justification:**

The explanation of the difference in theoretical analysis compared to prior work makes sense to me. Therefore, I modified the assessment of significance, originality, and the final score.

**Limitations:**

Yes

**Quality:**

3

**Strengths And Weaknesses:**

Strengths:
- Clear explanation of the convergence analysis. I appreciate the completeness but I do think some of the proofs can be deferred to the appendix.
- New setting in simple bilevel optimization problems where the lower-level objective is non-convex. To my knowledge, the general bilevel optimization problems with non-strongly-convex objective have been shown to have some impossibility results. It is good to see that the simple bilevel optimization problems can at least achieve some convergence result.

Weaknesses:
- Weak baselines in the experiments. Fixed penalty method is known to be not working.
- Unclear lower bound for the convergence rate. It would be great if the authors can establish some lower bound / impossibility results for the non-convex simple bilevel optimization problems.
- Most of the analyses are a direct application of DBGD by Gong et al. including the algorithm. The major difference between Gong et al.'s paper and this paper is just the problem setting. Gong et al. considers single level optimization with constraints, and this paper considers simple bilevel optimization that can be reformulated as constrained optimization by the value-based method. The technical novelty is weak.

---

> ### Author Rebuttal · Authors · 2025-07-31
>
> **W1. Weak baselines in the experiments. Fixed penalty method is known to be not working.**
>
> **R1.**
> Thank you for your comment. We would like to note that nonconvex simple bilevel optimization remains largely underexplored, and to the best of our knowledge, no existing discrete-time algorithm is specifically designed for this setting with theoretical guarantees for both levels.
>
> The effectiveness of the fixed penalty method in the nonconvex simple bilevel setting is still unclear. However, since it has established theoretical guarantees in the convex simple bilevel setting [1, 2], we include it as a baseline in our experiments. Notably, the original DBGD paper [3] also adopts the fixed penalty method as its only baseline.
>
> While there are no existing methods specifically tailored to the nonconvex simple bilevel setting, we include two additional baselines—BigSAM [3] and a-IRG [4]—which were originally developed for the convex case. Below we briefly describe the update rules of these algorithms.
>
> **BigSAM** is given by:
>
> $$
> y_{k+1} = x_k - \eta_g \nabla g(x_k),
> $$
>
> $$
> z_{k+1} = x_k - \eta_f \nabla f(x_k)
> $$
>
> $$
> x_{k+1} = \alpha_{k+1} z_{k+1} + (1 - \alpha_{k+1}) y_{k+1},
> $$
>
> where $\eta_f$ and $\eta_g$ are constant stepsizes, and $\alpha_k = \min(\frac{\gamma}{k}, 1)$ for some $\gamma > 0$.
>
> **a-IRG** is given by:
>
> $$
> \mathbf{x}_{k+1} =  \mathbf{x}_k - \gamma_k (\nabla g(\mathbf{x}_k) + \eta_k \nabla f(\mathbf{x}_k)),
> $$
>
> where $\gamma_k = \frac{\gamma_0}{\sqrt{k + 1}}$ and $\eta_k = \frac{\eta_0}{(k + 1)^{1/4}}$ for some constants $\gamma_0$ and $\eta_0$.
>
> Following the same setup as in our original paper, we report the final gradient norms and $\cos \theta$ after 1000 iterations. The table below summarizes the performance of the considered algorithms in the first experiment, with parameters chosen via grid search.
>
> | **Method**            | **Final $\|\nabla g\|^2$**       | **Final $\|\nabla_{\perp} f\|^2$**     | **Final $\cos(\theta)$** |
> |-----------------------|----------------------------------|----------------------------------------|---------------------------|
> | DBGD                  | $8.5657 \\times 10^{-17}$         | $1.0596 \times 10^{-16}$               | $-1.0000$                 |
> | Penalty $\lambda=1$   | $1.8142$                          | $4.4409 \times 10^{-16}$               | $1.0000$                  |
> | Penalty $\lambda=10$  | $2.4473 \times 10^{-1}$           | $1.9800 \times 10^{1}$                 | $0.2813$                  |
> | Penalty $\lambda=100$ | $6.3210 \times 10^{-3}$           | $9.4788$                               | $0.8692$                  |
> | Penalty $\lambda=1000$| $7.2475 \times 10^{-5}$           | $1.7477 \times 10^{1}$                 | $0.7334$                  |
> | BigSAM                | $2.7177 \times 10^{-4}$           | $2.1026 \times 10^{1}$                 | $0.4741$                  |
> | a-IRG                 | $9.3776 \times 10^{-7}$           | $1.9802 \times 10^{1}$                 | $0.6903$                  |
>
> *Table: Toy Example*
>
> For the second experiment, we present results using plots generated with various parameter settings in our original paper. Since images cannot be included in the rebuttal, we instead report the averaged results over these parameter settings for each method in the tables below. The total number of iterations is set to $10^6$.
>
> | **Method** | $\|\nabla g\|$         | $\|\nabla_{\perp} f\|$ | $g(U)$              | $f(U)$     |
> |------------|------------------------|-------------------------|---------------------|------------|
> | DBGD       | $5.59 \times 10^{-3}$  | $4.72 \times 10^{-1}$   | $2.73 \times 10^{-7}$| $129.32$   |
> | Penalty    | $9.79 \times 10^{-1}$  | $1.06$                  | $1.82 \times 10^{-2}$| $134.67$   |
> | BigSAM     | $4.54 \times 10^{-3}$  | $5.71$                  | $3.96 \times 10^{-4}$| $134.80$   |
> | a-IRG      | $1.89 \times 10^{-2}$  | $1.81$                  | $1.30 \times 10^{-4}$| $135.40$   |
>
> *Table: Matrix Factorization $f_1$*
>
> | **Method** | $\|\nabla g\|$         | $\|\nabla_{\perp} f\|$ | $g(U)$              | $f(U)$     |
> |------------|------------------------|-------------------------|---------------------|------------|
> | DBGD       | $7.12 \times 10^{-3}$  | $3.95 \times 10^{-1}$   | $8.15 \times 10^{-7}$| $37.042$   |
> | Penalty    | $1.109$                | $2.23$                  | $3.57 \times 10^{-2}$| $48.543$   |
> | BigSAM     | $4.55 \times 10^{-3}$  | $7.72$                  | $3.96 \times 10^{-4}$| $51.254$   |
> | a-IRG      | $2.43 \times 10^{-2}$  | $2.75$                  | $1.05 \times 10^{-4}$| $52.709$   |
>
> *Table: Matrix Factorization $f_2$*
>
> As shown in the table, it is not surprising that BigSAM and a-IRG underperform compared to DBGD in terms of our proposed stationarity metrics, as they are not specifically designed for the nonconvex setting. In particular, their performance is similar to that of the penalty method with a large penalty parameter—overemphasizing the lower-level objective while failing to adequately control the upper-level. The failure of algorithms designed for convex simple bilevel optimization when applied to nonconvex simple bilevel problems highlights the necessity of studying the nonconvex setting.
>
> ---
>
> **W2. Unclear lower bound for the convergence rate. It would be great if the authors can establish some lower bound / impossibility results for the non-convex simple bilevel optimization problems.**
>
> **R2.** We appreciate this insightful comment. However, we note that nonconvex simple bilevel optimization is still in its early stages of study. Our work provides the first upper-bound complexity result for a discrete-time algorithm that guarantees joint stationarity for both levels in nonconvex simple bilevel problems.
>
> Establishing lower bounds or impossibility results is indeed an important and interesting direction. While focusing solely on the lower-level or upper-level problem can achieve a rate of $\mathcal{O}(\epsilon^{-2})$, in the simple bilevel setting, we must consider both objectives jointly. This introduces a trade-off between the convergence rates of the lower- and upper-level objectives, making the construction of lower bounds fundamentally different from the previous lower bound results and substantially more challenging.
>
> ---
>
> **W3.
> Most of the analyses are a direct application of DBGD by Gong et al. including the algorithm. The major difference between Gong et al.'s paper and this paper is just the problem setting. Gong et al. considers single level optimization with constraints, and this paper considers simple bilevel optimization that can be reformulated as constrained optimization by the value-based method. The technical novelty is weak.**
>
> **R3.**
> While there are some similarities between the algorithm analyzed in our paper and the DBGD method proposed in [3], we would like to emphasize that our analytical framework and convergence results differ substantially from those in [3]. Below, we explicitly highlight the key differences:
>
> **(1)** In our work, we first introduce a notion of stationarity for nonconvex simple bilevel optimization by decomposing the upper-level gradient into two orthogonal components: one parallel to the lower-level gradient and the other orthogonal to it. This definition admits both an intuitive and rigorous interpretation, analogous to the concept of stationarity in single-level optimization, as demonstrated in Lemma 3.1. Additionally, we explore the connection between our proposed stationarity metric and existing metrics used in constrained optimization and general bilevel optimization. None of these are addressed in [3].
>
> **(2)** In the first part of their results (Section 3.2), [3] assumes that $\lambda_k$ is uniformly upper-bounded (Assumption 3.4) in the constrained optimization setting, which is not possible in simple bilevel optimization. This assumption also simplifies their analysis substantially, as we noted in line 260.
>
> **(3)** In the second part of their results (Section 3.3), as mentioned in Remark 5.2, [3] analyzes the continuous-time limit of the algorithm. However, their analysis does not account for the additional error arising from approximating $f$ and $g$ using first-order Taylor expansions. Consequently, their convergence result cannot be directly translated into a concrete bound for the discrete-time algorithm. In contrast, our contributions include explicitly addressing this discretization error—requiring the solution of an implicit inequality (cf. Lemma 5.3), careful selection of the step size $\eta$ and hyperparameter $\beta$, as well as removing the common assumption of uniformly bounded $\\|\nabla g\\|$.
>
> **(4)** Regarding convergence guarantees, Proposition 3.8 in [3] relies on the constant rank constraint qualification (CRCQ), which is known to be difficult to satisfy in bilevel optimization, as discussed in [6]. In contrast, our analysis does not depend on this assumption, as we use a different stationarity metric. For our proposed metric, no additional non-standard assumptions are required. To relate our metric to the one used in [3], an additional local error bound condition is needed—but not the more restrictive CRCQ assumption, as demonstrated in Theorem 3.2.
>
> ---
>
> References:
>
> [1]. Samadi et al. Achieving optimal complexity guarantees for a class of bilevel convex optimization problems.
>
> [2]. Chen et al. Penalty-based methods for simple bilevel optimization under h\"{o} lderian error bounds.
>
> [3]. Gong et al. Bi-objective trade-off with dynamic barrier gradient descent.
>
> [4]. Sabach et al. A first-order method for solving convex bilevel optimization problems.
>
> [5]. Kaushik et al. A method with convergence rates for optimization problems with variational inequality constraints.
>
> [6]. Xiao et al. A Generalized Alternating Method for Bilevel Learning under the Polyak-{\L} ojasiewicz Condition.

---

> > ### Comment · Reviewer_oo1n · 2025-08-04
> >
> > Thank you for your thorough explanation and the additional numerical experiments. The response mostly addresses my concerns. I agree that there are not many existing algorithms for simple bilevel optimization for comparison and the theoretical analysis is legit. With that, I am happy to raise the score.

---

### Decision · Program_Chairs · 2025-09-17

**Decision:**

Accept (poster)

**Comment:**

The paper focuses on solving bilevel optimization problems, where the upper-level objective is minimized over the solution set of the lower-level problem. The case where both objectives might be nonconvex is examined. Since the problem is computationally intractable in that regime, the authors introduce a new (reasonable) notion of stationarity and show that a variant of Dynamic Barrier GD achieves approximate stationarity. The paper received mixed reviews but during the rebuttal, the authors provided thorough explanations and made all reviewers having positive or slightly positive opinion. We recommend acceptance and advise the authors to invoke the feedback from the reviewers to their camera ready version.